# Randomized controlled trial of molnupiravir SARS-CoV-2 viral and antibody response in at-risk adult outpatients

Viral clearance, antibody response and the mutagenic effect of molnupiravir has not been elucidated in at-risk populations. Non-hospitalised participants within 5 days of SARS-CoV-2 symptoms randomised to receive molnupiravir (n = 253) or Usual Care (n = 324) were recruited to study viral and antibody dynamics and the effect of molnupiravir on viral whole genome sequence from 1437 viral genomes. Molnupiravir accelerates viral load decline, but virus is detectable by Day 5 in most cases. At Day 14 (9 days post-treatment), molnupiravir is associated with significantly higher viral persistence and significantly lower anti-SARS-CoV-2 spike antibody titres compared to Usual Care. Serial sequencing reveals increased mutagenesis with molnupiravir treatment. Persistence of detectable viral RNA at Day 14 in the molnupiravir group is associated with higher transition mutations following treatment cessation. Viral viability at Day 14 is similar in both groups with post-molnupiravir treated samples cultured up to 9 days post cessation of treatment. The current 5-day molnupiravir course is too short. Longer courses should be tested to reduce the risk of potentially transmissible molnupiravir-mutated variants being generated. Trial registration: ISRCTN30448031

Treatment of SARS-CoV-2 with the nucleoside analogue molnupiravir (MK4482, EIDD2801) was reported to reduce viral load, hospitalisation and mortality in unvaccinated participants with early COVID-19 in the MOVeOUT trial[1,2]. Based on these data, molnupiravir received emergency use authorisation in the UK in November 2021 for early treatment of SARS-CoV-2 in individuals deemed to be at higher risk of complications due to age or underlying comorbidities.

Molnupiravir is metabolised intracellularly to NHC-triphosphate, which competes with natural cytidine and uridine for incorporation by the viral RNA-dependent RNA polymerase (RdRp) into the nascent viral RNA. This leads to abnormal, non-Watson-Crick pairing with guanosine and uridine in further rounds of replication, increasing the substitution of adenosine for guanosine and cytosine for uridine, so-called transition mutations, within the SARS-CoV-2 genome. Lethal mutagenesis resulting from treatment with RdRp inhibitors eventually leads to viral extinction[3,4]. A distinctive pattern of transition mutagenesis is evident in viral genomes recovered from animals and humans who have received molnupiravir[3–5]. The risk that, following molnupiravir treatment, some highly mutated viruses might remain viable and capable of onward transmission has been postulated[6,7].

To measure the impact of molnupiravir in a largely vaccinated population, the Platform Adaptive trial of NOvel antiviRals for eArly treatMent of covid-19 In the Community (PANORAMIC) was established. The first drug tested in PANORAMIC was molnupiravir, and amongst 25,783 mostly vaccinated individuals, found that molnupiravir did not reduce hospitalisation or death[8] (primary endpoint). Secondary outcomes showed those receiving molnupiravir experienced significantly reduced viral load during treatment and reported faster symptom recovery and fewer general practitioner consultations than those receiving Usual Care.

Here we report detailed results of the PANORAMIC[8] virology substudy, where a subset of participants in both arms underwent serial virology and immunology sampling. Demographic, clinical, and viral load data together with biomarkers of immune response (anti-SARS-

✉e-mail: j.standing@ucl.ac.uk

CoV-2 spike antibody) and disease severity (high sensitivity C-reactive protein (CRP)) were collected to study viral and immune dynamics. SARS-CoV-2 genome sequencing and viral culture provide further insights into the risk-benefit profile of molnupiravir to patient and public health.

## Results

### Recruitment, demographics and baseline viral load, antibody, and CRP

Prior to the molnupiravir arm closing, 657 out of 6127 participants approached agreed to take part in the virology sub-study, with 94 participants sent kits for intensive sampling (daily nasopharyngeal swab for 7 days plus Day 14) and 563 for less intensive sampling (nasopharyngeal swab on Days 1, 5 and 14). A recruitment flow chart is provided in Fig. 1. All participants were asked to provide a dried blood spot on Days 1, 5 and 14. The Day 1 swab and blood spot were performed prior to treatment commencing and so constituted the baseline sample. Of these 81 (86%) intensively sampled participants and 500 (89%) less intensively sampled participants returned swabs. Overall, 2014/2441 (82.5%) swabs and 1608/1731 (92.9%) dried blood spot samples were returned. Four participants were excluded due to all swabs being undetectable for virus, yielding a final cohort of 577 participants with 1990 viral loads and, after removing 25 dry blood spot samples with insufficient material for analysis and those from the four excluded participants, 1566 spike antibody measures were retained. More participants in Usual Care enroled, but number of analysable samples returned per participant, baseline demographics, viral load, spike antibody and days since symptom onset were balanced between groups (Table 1). There was however a significantly higher proportion of female participants in the Usual Care arm.

Baseline viral loads increased with age, decreased with time since symptom onset, and were negatively correlated with baseline spike antibody and positively correlated with baseline capillary CRP (Fig. 2a–d). Males had higher baseline viral load, but otherwise, apart

from higher viral loads in the small number ($n = 8$) of participants with kidney disease, no correlations were found with comorbidities, vaccine doses, or receipt of inhaled corticosteroids (Fig. 2e). Baseline spike antibodies were higher in female participants and lower in participants who were not fully vaccinated but not correlated with co-morbidities (Fig. 2f). Participants in the molnupiravir arm were asked how many days of the 5-day treatment course they completed. Only 10 participants reported not taking the full course equating to 4% of the treated arm. Of these, 5 participants reporting not taking any, two only 1 day and one each reporting 2, 3 or 4 days of treatment.

### Viral load and spike antibody dynamics with and without molnupiravir treatment

Viral load declined significantly faster in molnupiravir-treated participants during treatment, but at the end of the 5-day course, only a minority (14%) had viral load below the lower limit of quantification (<LLOQ) (Fig. 3a). In the multivariable viral dynamic model, sex, age, time since symptom onset and baseline spike antibody titre were significant predictors of baseline viral load (Table 2, Supplementary Fig. 1 and 2). No covariate affected the slope of viral load decline except for molnupiravir. Molnupiravir significantly increased viral decline rate during treatment but significantly decreased it following treatment cessation (Fig. 3a, b). The model predicts male participants have on average 0.2 log10 copies/mL higher baseline viral load whereas for a 10 year increase in age from the median a 0.13 log10 copies/mL increase in viral load is expected. A 0.5 log10 U/mL decrease in spike antibody from the population median is associated with a 0.18 log10 copies/mL increase in viral load. Time since symptom onset increasing from 2 to 5 days was associated with a 0.9 log10 copies/mL decrease in viral load. The viral decline half-life in Usual Care was 0.72 days. This decreased to 0.41 days when on molnupiravir, and then increased to 1.71 days post-treatment.

By Day 14 viral loads were significantly higher in molnupiravir-treated participants than Usual Care. Extrapolation using simulated

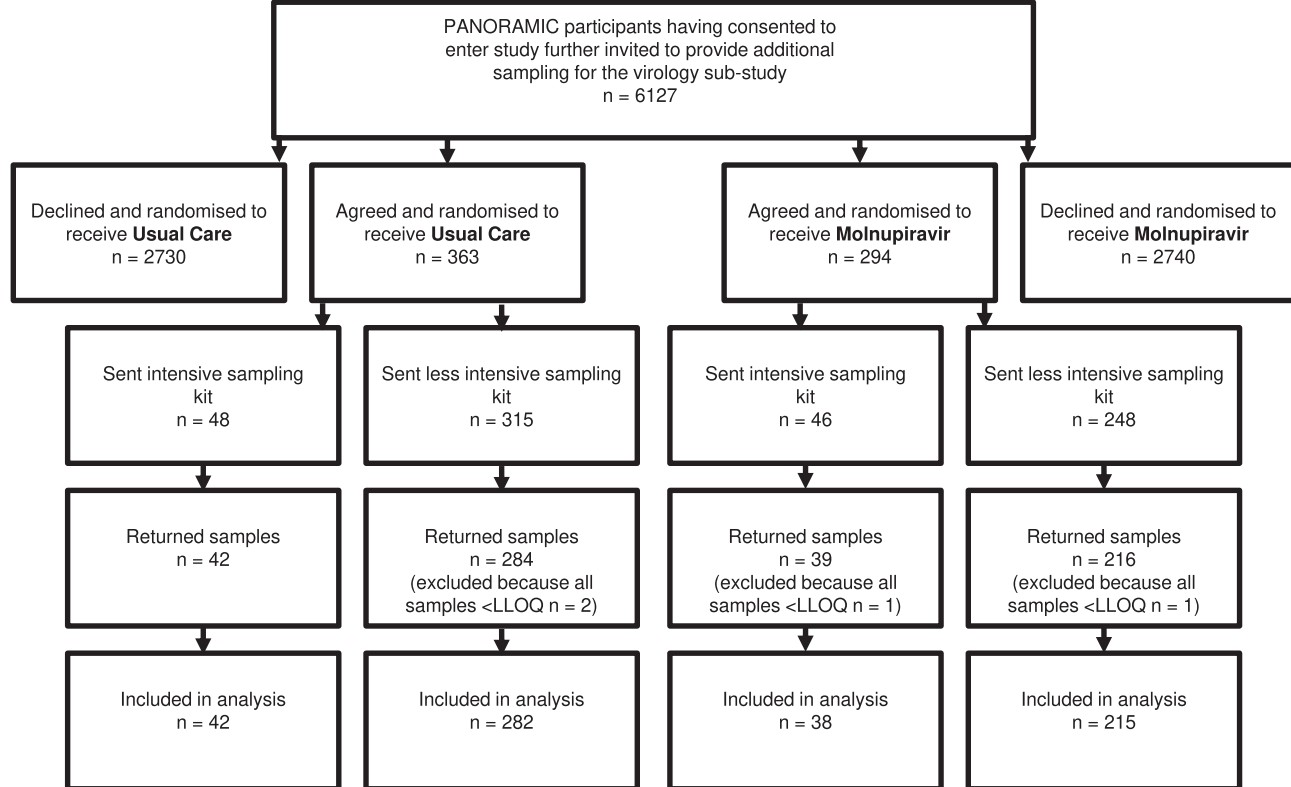

**Fig. 1 | Disposition of PANORAMIC participants approached and provided virology samples.** < LLOQ, below lower limit of quantification.

**Table 1 | Demographics for the included participants**

| Variable | Molnupiravir (n = 253) | Usual Care (n = 324) | P-value |
|---|---|---|---|
| Intensive sampling (%) | 38 (15%) | 42 (13%) | 0.557 |
| Median number of viral load measurements per participant intensive group (range) | 8 (4,8) | 8 (4,8) | – |
| Median number of viral load measurements per participant less-intensive group (range) | 3 (1,3) | 3 (1,3) | – |
| Median number of spike antibody measurements per participant (range) | 3 (0,3) | 3 (0,3) | – |
| Age years (sd) | 58 (10.1) | 58 (10.9) | 0.844 |
| Female (%) | 139 (55%) | 212 (65%) | 0.0133 |
| Ethnicity non-white (%) | 7 (3%) | 9 (3%) | >0.999 |
| Fully vaccinated (≥3 doses) (%) | 240 (95%) | 314 (97%) | 0.3 |
| Inhaled corticosteroids (%) | 56 (22%) | 70 (22%) | 0.959 |
| Immune disease (%) | 23 (9%) | 21 (6%) | 0.311 |
| Obesity (%) | 43 (17%) | 46 (14%) | 0.419 |
| Any comorbidity (%) | 167 (66%) | 199 (61%) | 0.294 |
| PANORAMIC primary outcome: Hospitalised or died (%) | 2 (0.79%) | 3 (0.93%) | >0.999 |
| Mean days since symptom onset at baseline (sd) | 2.4 (0.78) | 2.5 (1.12) | 0.621 |
| Mean viral load (log10(cp/mL)) at baseline (sd) | 7.4 (1.14) | 7.3 (1.04) | 0.334 |
| Mean spike antibody (log10(U/mL)) at baseline (sd) | 3.3 (0.45) | 3.3 (0.44) | 0.91 |
| Mean capillary CRP (mg/L) at baseline (sd) | 0.68 (0.65) | 0.65 (0.64) | 0.544 |

Continuous variables are reported as mean and standard deviation apart from the number of measurements which are reported as mean (range). Categorical variables reported as number in each category and percentage. Comparisons between molnupiravir and usual care arm variables was made using a *t* test for continuous variables and a Chi square test for categorical variables

parameters from the viral dynamic model, but with the drug effect extended, suggest that to achieve a 95% probability of virus being <LLOQ, at least 10 days of therapy is required (Fig. 3c).

Anti-SARS-CoV-2 spike antibody concentration increased in both groups but was significantly lower by Day 14 in the molnupiravir group (Fig. 3d). In the multivariable antibody dynamic model, male sex and not being fully vaccinated correlated with low baseline spike antibody whereas only molnupiravir was a significant predictor of slope (Table 3, Supplementary Fig. 1b), with antibody doubling time significantly slower in molnupiravir-treated participants (Fig. 3e). The rate of increase in spike antibody was inversely correlated with area under the curve (AUC) of viral load, indicating virus exposure likely drives antibody production (Fig. 3f). The trajectory of CRP decline was similar in both groups (Supplementary Fig. 3).

**Molnupiravir associated viral genetic changes**

Of 1672 samples with viral loads above the LLOQ, 1436 generated usable genome sequences, defined as >90% coverage with >10X mean read depth. Sequencing success was strongly associated with viral load (Supplementary Fig. 4). The numbers of sequences generated per participant are shown in Supplementary Table 1. Samples were collected between March and April 2022 with lineage analysis assigning over 70% to clade Omicron BA.2 (72.9% in molnupiravir-treated participants and 75.2% in Usual Care; Supplementary Table 2).

In keeping with previous reports[8,9], increased mutations in molnupiravir-treated participants were observed, with a significantly increased ratio of transition (particularly G to A and C to T) to transversion mutations (*p* < 0.0001, Fig. 4a and Supplementary Fig. 6) and greater sequence diversity (Fig. 4b) as compared with Usual Care. The results were not affected by read depth (Supplementary Fig. 5). Mutations in both molnupiravir-treated and Usual Care participants were distributed across the genome (Supplementary Fig. 7), with Usual Care having fewer mutations overall (Fig. 4a).

Of 145 participants with viral Ct <38 at Day 14 (nine days following treatment end), 46/75 in the molnupiravir and 21/70 in the Usual Care group were successfully sequenced, the higher success rates in the molnupiravir samples likely due to higher viral loads. Molnupiravir-treated participants had higher numbers of mutations (Fig. 4a) and transition/transversion ratios (Supplementary Fig. 6) and on

phylogenetic analysis, longer branch lengths at Day 14 (Fig. 5a, b). Overall, molnupiravir-treated individuals with virus persisting until Day 14 had consistently higher viral diversity compared with those clearing before Day 14 (Fig. 4b).

Thirty two of 46 individuals in the molnupiravir-treated group had levels of drug-associated mutagenesis (as measured by transition fraction) of ≥90% at Day 14 (Supplementary Fig. 8). The remaining 13/45 had transition fractions of ≤89% and clustered with Usual Care. Sixty four of the 67 molnupiravir and Usual Care group with sequences on Day 14, including the 13 molnupiravir-treated participants with low transition fractions, also had sequence data available on Day 5. In 4/13 of those with low transition fractions on Day 14, their transition fraction on Day 5 also clustered with Usual Care, suggesting low drug exposure. The remaining 9/13 were indistinguishable from the other molnupiravir treated individuals (Supplementary Fig. 8). Data on self-reported adherence was uninformative, with all 13 participants reporting taking all doses. No baseline characteristics differed between these 13 and the remaining molnupiravir-treated participants (Supplementary Table 3).

In the multivariable model only treatment group was significantly associated with maximum mutation load, with molnupiravir being associated with on average 180 more mutations than Usual Care (*p* < 0.001). Similarly, only treatment group was associated with maximum transition to transversion ratio, with molnupiravir being associated with an average ratio of 9.6 versus 5.3 in Usual Care (*p* < 0.001). No significant associations were found with the appearance of spike escape mutations.

**Genetic changes provide insight into mechanism**

To investigate the potential functional impact of molnupiravir-associated mutations, we used the publicly available Pokay[10] database to identify post-baseline amino acid substitutions known to change SARS-CoV-2 spike protein function or increase escape from neutralisation by antibodies. Increased numbers of spike mutations-of-interest over time were identified at low (5–50%) and consensus level (>50%), particularly in molnupiravir-treated individuals, and more so in those with virus persisting beyond Day 5 (Supplementary Fig. 9). Emerging spike antibody escape or other mutations in molnupiravir-treated participants did not appear associated with viral decline

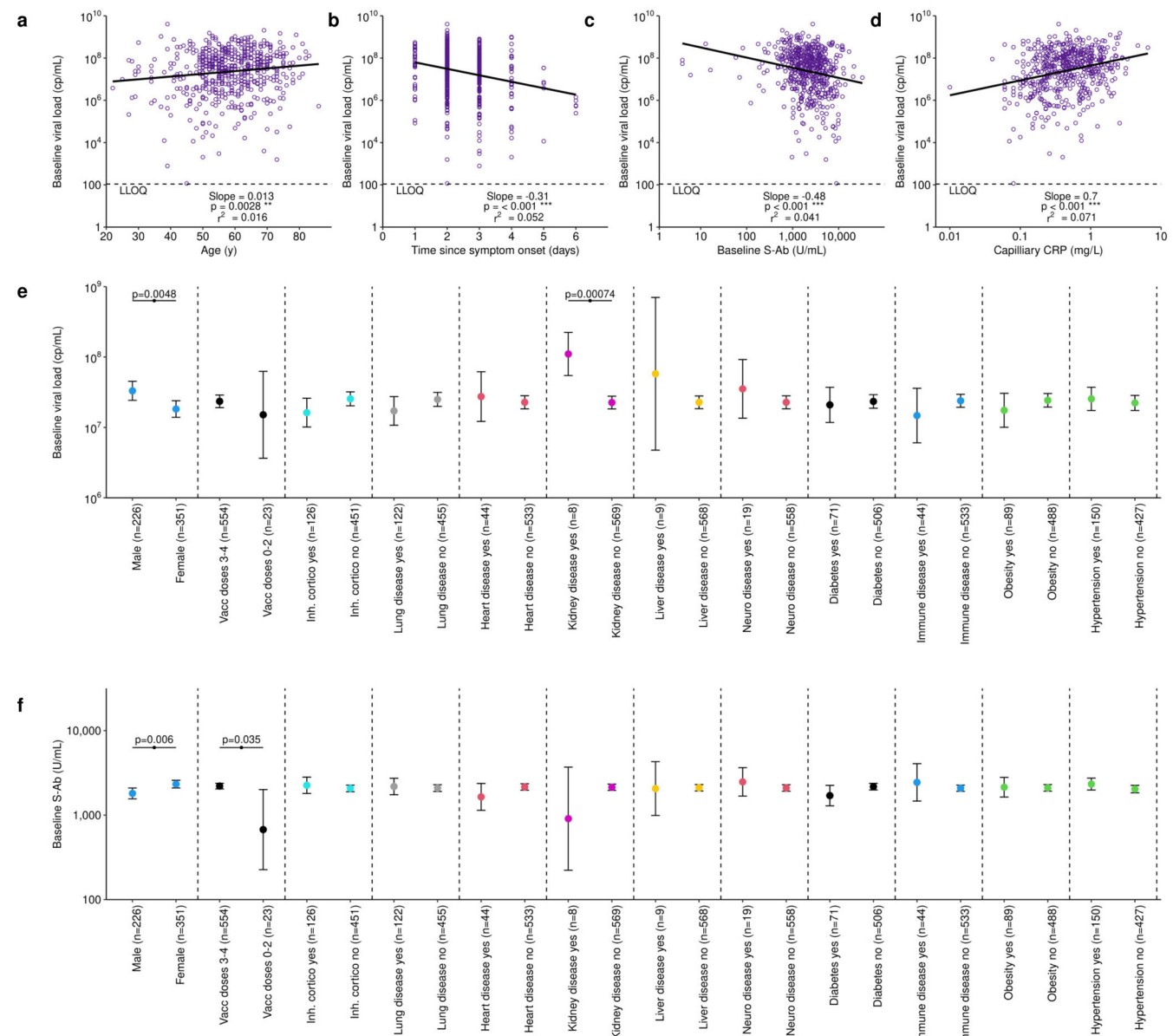

**Fig. 2 | Baseline viral load and spike antibody relationships with each other, CRP, demographics and co-morbidities. a** Baseline viral load increase with age. **b** Baseline viral load decrease with time since symptom onset. **c** Baseline viral load decrease with increased baseline spike antibody (S-Ab). **d** Baseline viral load increase with increase in CRP measured in capillary blood. **e** Geometric mean viral load ±95%CI with time compared with a *t* test showing baseline viral load higher in males and in patients with kidney disease. **f** Geometric mean S-Ab ±95%CI with time compared with a *t* test showing baseline S-Ab lower in males and participants who were not fully vaccinated. LLOQ is lower limit of quantification which for viral load was 109 cp/mL. For (**a**–**d**) the slope was compared with zero using the f statistic, for figures (**e**) and (**f**) and *t* test was used to compare geometric mean values. *$P < 0.05$, **$P < 0.01$, ***$P < 0.001$. Source data are provided as a Source Data file.

(Fig. 4b and Supplementary Fig. 10). We also used the publicly available ResistanceDB[11] molnupiravir-resistance mutations based on the literature. We identified one consensus and one low level RdRp mutation predicted to cause resistance (Supplementary Table 4).

At Day 14, nine new amino acid substitutions were present at consensus level (>50%) in sequences from two or more participants taking molnupiravir and one in Usual Care (Fig. 6c). Two of the recurrent post-baseline consensus substitutions, present only in molnupiravir-treated participants, were in the polymerase (NSP 12) and the accessory protein NSP8, associated with viral replication (Fig. 6c). A third substitution, F694Y, in NSP12 has been previously identified as an artefact of the ARTIC primer amplification and was therefore excluded[12]. The recurrent NSP8 mutation is uncommon among UK viruses (0.01%) in the GISAID database and was present at a low level (<6%) in three molnupiravir samples (Day 5) taken during this study.

The mutation is not known to be in a position that affects RdRp function. The recurrent RdRp substitution N507I in Usual Care was present in 26 participants, nine in the molnupiravir-treated and 17 in Usual Care, including at baseline. Nine of the 26 N507I substitutions occurred at consensus level, the majority in viruses from participants with <LLOQ viral loads on Day 14.

In contrast, the A716V RdRp (NSP12) substitution, although not known to be a molnupiravir-resistance mutation, is within the catalytic site (Supplementary Fig. 11). A716V was absent in Usual Care but present at <10% allele frequency during or just after treatment in a further six molnupiravir-treated participants, none of whom had detectable viral loads at Day 14. The only recurrent Day 14 amino acid change present in the spike protein, F490S, occurred in molnupiravir-treated participants and is in the receptor binding domain (RBD).

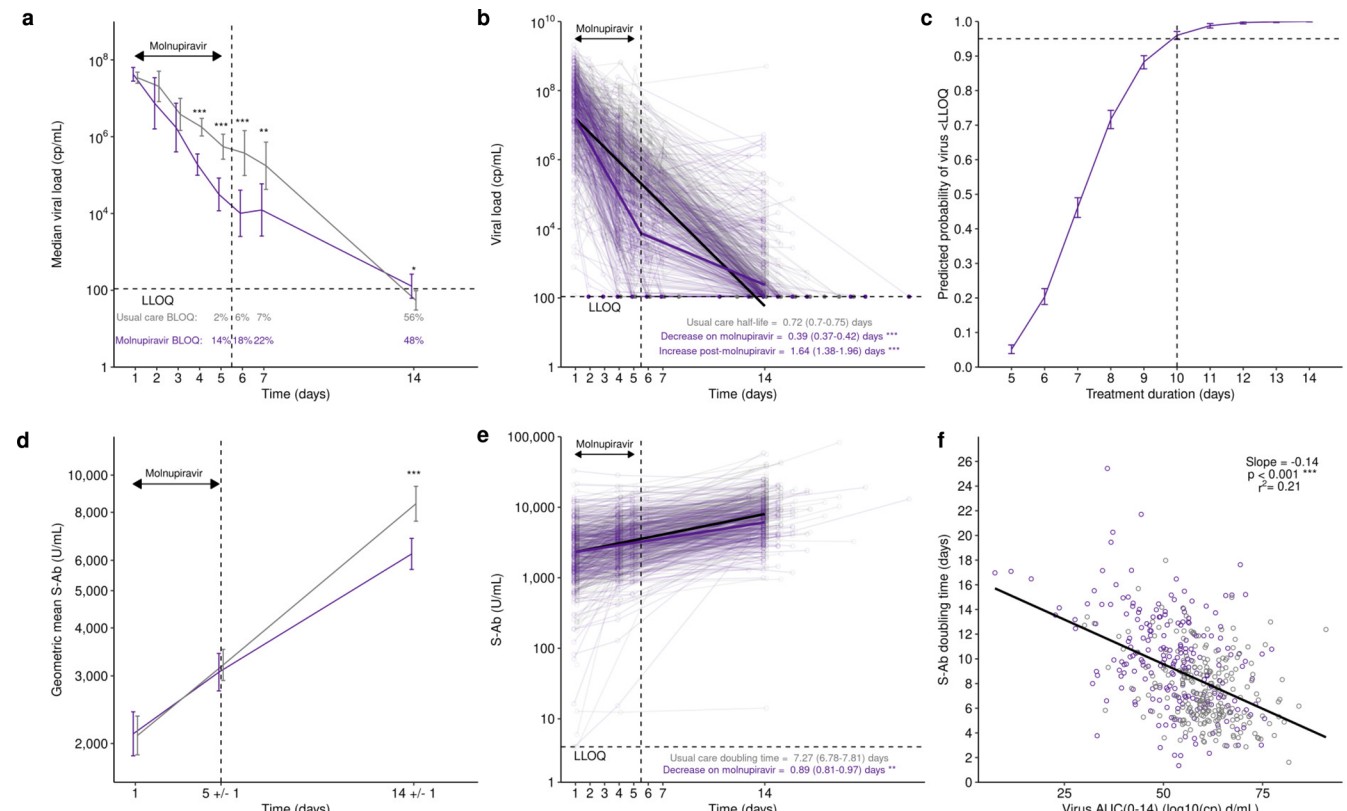

**Fig. 3 | Viral and spike antibody dynamics in molnupiravir (purple) and Usual Care (grey) arms. a** Median viral load ±95%CI with time, below limit of quantification (BLOQ) measures substituted with LOQ/2 for molnupiravir (*n* = 253) and Usual Care (*n* = 324) compared with a Mann–Whitney test. **b** Viral load with time showing multivariable linear viral dynamic model predictions for a typical individual (female, median age, S-Ab and time since symptom onset) receiving molnupiravir (purple line) or Usual Care (black line). For molnupiravir a piecewise increase then decrease in elimination rate was estimated with a change point at the end of treatment compared with a likelihood ratio test. **c** Simulation from the viral dynamics model for 1000 subject demographic combinations sampled from the

data. Line shows the 50th percentile of the probability of viral load being <109 cp/mL on the last day of treatment for 5 to 14 days treatment. Error bars are 95% prediction intervals. **d** Geometric mean spike antibody ±95%CI with time compared with a *t* test. **e** Spike antibody with time showing multivariable linear antibody dynamic model predictions for a typical individual (female, median age) receiving molnupiravir (purple line) or Usual Care (black line) compared with a likelihood ratio test. **f** Model-derived individual predictions of spike antibody doubling time versus viral load area under the curve (AUC) from day 0 to 14. *$P < 0.05$, **$P < 0.01$, ***$P < 0.001$. Source data are provided as a Source Data file.

## Table 2 | Parameter estimates for the viral dynamic model

| Parameter | Estimate (95%CI) | IIV (%CV) |
|---|---|---|
| δ (d⁻¹) | 0.96 (0.93, 0.993) | 12.4 |
| V(0) (log10 cp/mL) | 7.189 (7.081, 7.298) | 74.7 |
| Additive error (log10 cp/mL) | 1.002 | – |
| $\beta_{mol\_on}$ | 1.766 (1.653, 1.888) | – |
| $\beta_{mol\_off}$ | 0.422 (0.354, 0.502) | – |
| $\beta_{sex}$ | 1.248 (1.113, 1.398) | – |
| $\beta_{age}$ | 0.833 (0.492, 1.173) | – |
| $\beta_{ab}$ | −1.109 (−1.622, −0.654) | – |
| $\beta_{sym}$ | −0.870 (−1.113, −0.626) | – |

The slope, δ, is the first-order decline rate in viral load, whereas the intercept (V(0)) the baseline viral load. Inter-individual variability (IIV) was assumed to follow a log-normal distribution for δ and log10 for V(0). Additive error is the variance of the residual error term. The following covariates all significantly (p < 0.01) improved model fit: $\beta_{mol\_on}$ refers to the fractional change in delta when on molnupiravir, $\beta_{mol\_off}$ refers to the fractional change in delta in the post-molnupiravir treatment period. $\beta_{sex}$ is the change in baseline viral load in males. $\beta_{age}$ is the allometric exponent relating age with V(0). $\beta_{ab}$ is the allometric exponent for the inverse relationship of V(0) load and spike antibody. $\beta_{sym}$ is the allometric exponent for the inverse relationship between V(0) and time since symptom onset. The final model describing viral load with time (V(t)) was therefore: $V(t) = V(0)\beta_{sex}\,I_{male}(age_i/age_{median})^{\beta_{age}}(A(0)_i/A(0)_{median})^{\beta_{ab}}(tsym_i/tsym_{median})^{\beta_{sym}}exp\{-\delta\,t\,I_{mol1}\beta_{mol\_on}I_{mol2}\beta_{mol\_off}\}$ where $I_{male}$ is an indicator for male participants; age, A(0) and tsym are the individual and population median age in years, log10 spike antibody baseline and tsym time since symptom onset respectively; $I_{mol1}$ and $I_{mol2}$ are time-varying indicators for the molnupiravir arm during and post-treatment.

To further examine the potential functional significance of mutations in viral persistence, we used the Pokay database[10] to identify non-synonymous spike-escape mutations present at consensus level in Day 14 sequences (Supplementary Table 4). None of the five

participants with either a consensus-level spike antibody escape mutation or a recurrent spike mutation (F490S) at Day 14 had the same substitution prior to Day 14. Other recurrent consensus mutations present in ORFs1a and 3a are of unknown significance and in certain

cases, for example A99T, frequently described[12]. S1952L and P104S were also present at baseline in one and two participants respectively.

## Viral culture

Persisting viral RNA does not equate to infectious virions and previous studies have demonstrated that following molnupiravir treatment viral fitness measured through tissue culture may be impaired[13]. We collected swabs in viral transport medium which were cultured with Calu-

**Table 3 | Parameter estimates for the spike antibody dynamic model**

| Parameter | Estimate (95%CI) | IIV (%CV) |
|---|---|---|
| δ (d⁻¹) | 0.095 (0.089, 0.102) | 58.2 |
| A(0) (log10 U/mL) | 3.365 (3.331, 3.399) | 39.3 |
| Additive error (log10 U/mL) | 0.164 | – |
| $\beta_{sex}$ | 0.942 (0.886, 1.001) | – |
| $\beta_{vac}$ | 0.772 (0.608, 0.979) | – |
| $\beta_{mol}$ | 0.781 (0.712, 0.856) | – |

The slope, δ, is the first-order increase rate in spike antibody, whereas the intercept (A(0)) the baseline spike antibody level. Inter-individual variability (IIV) was assumed to follow a log-normal distribution and estimated for δ and A(0). Additive error is the variance of the residual error term. The following covariates all significantly ($p < 0.01$) improved model fit: $\beta_{sex}$ is the fractional change in A(0) in males. $\beta_{vac}$ is the fractional change in A(0) in participants who were not full vaccinated. $\beta_{mol}$ is the fractional change in δ in participants receiving molnupiravir. The final model describing antibody with time (A(t)) was therefore:

$A(t) = A(0)\ \beta_{sex} I_{male}\ \beta_{vac} I_{unvacc}\ \exp\{- \delta\ t\ I_{mol}\ \beta_{mol}\}$,

where $I_{male}$ is an indicator for male participants, $I_{unvacc}$ is an indicator for participants who were not fully vaccinated, and $I_{mol}$ is an indicator for participants who took molnupiravir.

3 cells. Using a high throughput culture method with screening over 7 days for evidence of cytopathic effect and the presence of SARS CoV-2 by lateral flow immunochromatography and PCR, we were able to recover viable virus from 16.7% of all tested samples including baseline. The positive culture rate for samples collected during treatment (Days 2-5) was 10.4% with molnupiravir versus 15.2% for Usual Care. Post-treatment viability (Days 6-20) dropped to 5.1% for molnupiravir and 2.5% for Usual Care. Positive culture was associated with higher viral load at time of sampling (Supplementary Fig. 12a) but not obviously affected by transit time by post after sample collection (Supplementary Fig. 12b).

Over 43% of baseline samples were culture positive, and viral recovery subsequently declined with time for both Usual Care and molnupiravir samples (Extended Data Fig. 12c). Overall, however, we did not find a difference in recoverable virus between groups (Fig. 7). Post-treatment samples from six molnupiravir participants were culturable.

## Discussion

This first report of a PANORAMIC virology-sub-study has shown that important virology and immunology insights are feasible through minimally invasive participant self-sampling. In the main trial molnupiravir treatment did not demonstrate benefit in reducing hospitalisations or deaths compared to Usual Care[8]. We now show that while 800 mg molnupiravir twice daily for 5 days causes initial significant viral load reductions, treatment is independently associated with increased risk of detectable virus at Day 14 likely resulting from too short a treatment duration. The residual virus in participants receiving

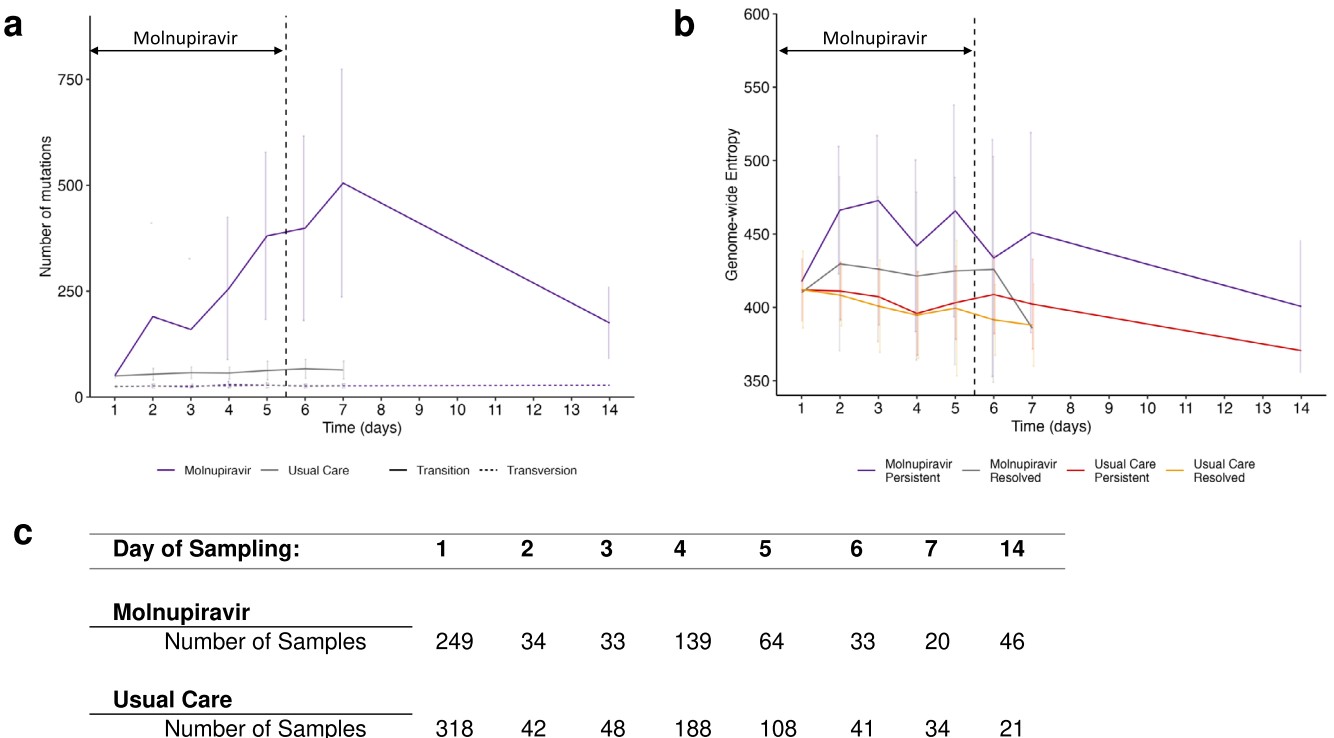

**Fig. 4 | Mutations occurring over time in viral sequences. a** Number of transition and transversion mutations in molnupiravir-treated (purple) and Usual Care (grey) participants. Transition mutations (G to A, A to G, C to T and T to C) are shown as solid lines, transversions (G to C, C to G, A to T, C to A, G to T, T to G and T to A) are shown as dotted lines; standard deviation denoted by error bars. **b** Shannon entropy over time in participants receiving molnupiravir or Usual Care. Participants in each group with viral loads above the LLOQ by day 14 (Molnupiravir Persistent; purple and Usual Care Persistent; orange) and participants with <LLOQ viral loads by Day 14 (Molnupiravir Resolved; grey or Usual Care Resolved; yellow) are shown; standard deviation denoted by error bars. **b** Shannon entropy over time in participants receiving molnupiravir or Usual Care. Participants in each group with viral loads above the LLOQ by day 14 (Molnupiravir Persistent; purple and Usual Care Persistent; orange) and participants with <LLOQ viral loads by Day 14 (Molnupiravir Resolved; grey or Usual Care Resolved; yellow) are shown. **c** Numbers of sequenced samples at each time point. Source data are provided as a Source Data file.

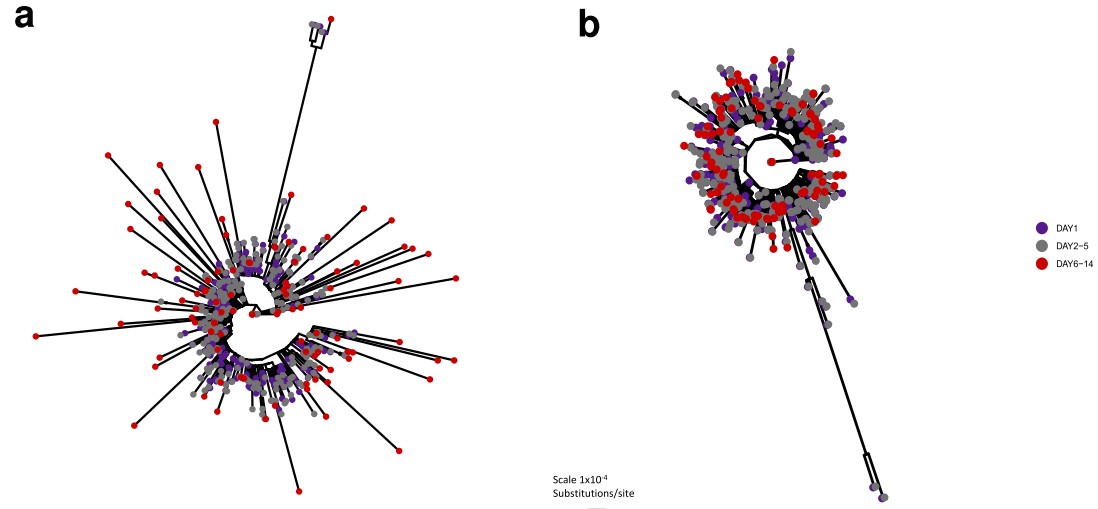

**Fig. 5 | Maximum likelihood phylogenetic trees of consensus sequences from sequenced samples. a** sequences (*n* = 627) from molnupiravir treated participants. **b** sequences (i = 809) from participants receiving Usual Care. Baseline samples (Day 1) are shown in purple, samples collected between days two and five (corresponding to the duration of molnupiravir treatment) are shown in grey, samples collected between Days 6 and 14 (after treatment has finished in the molnupiravir arm) are shown in red. The scale bar is shown.

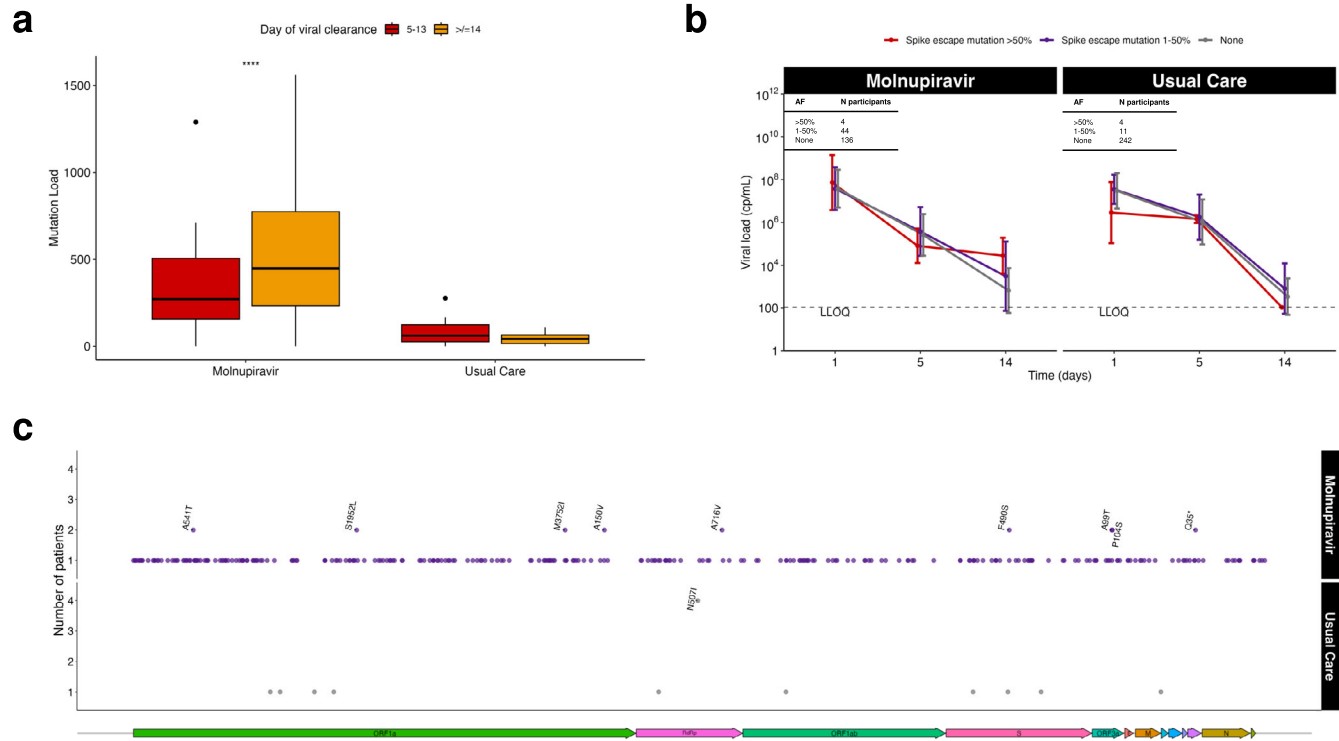

**Fig. 6 | Viral mutations in molnupiravir and Usual Care groups. a** The number of mutations (mean= 553 and standard deviation=391), in samples taken at treatment-end (Day 5) is significantly higher (Mann–Whitney test) in molnupiravir-treated participants with detectable virus at Day 14 (orange, *n* = 100 participants) compared with those who cleared virus between Days 5–14 (red, *n* = 87 participants). Box-plot midline is the median, the upper and lower limits the 75th and 25th percentile, the whiskers 1.5 times the interquartile range. Data beyond that are represented as points. There is no difference between the numbers of mutations in the comparable Usual Care groups (mean = 45 and standard deviation=31.2, *n* orange = 94 participants, *n* red = 158 participants). **b** Viral load trajectories (mean and standard deviation) for molnupiravir-treated and Usual Care participants with any know spike neutralisation-escape missense mutation occurring at any time point post baseline at consensus level (allele frequency (AF) > 50%) in red, below consensus level (1–50%) in purple or with no mutations (grey). Participants (441) were required to have viral load data for baseline, Day 5 and Day 14 and sequence data for all positive viral loads. The table shows the total number of participants with consensus, below consensus and zero known neutralisation escape mutations for molnupiravir and Usual Care groups, **c** Consensus level (>50%) amino acid substitutions occurring in Day 14 sequences from molnupiravir (purple) and Usual Care (grey) participants. Mutations occurring in more than one participant's sample are shown above the rest. Schematic of the SARS-CoV-2 genome is shown along the bottom. *****P* < 0.0001. Source data are provided as a Source Data file.

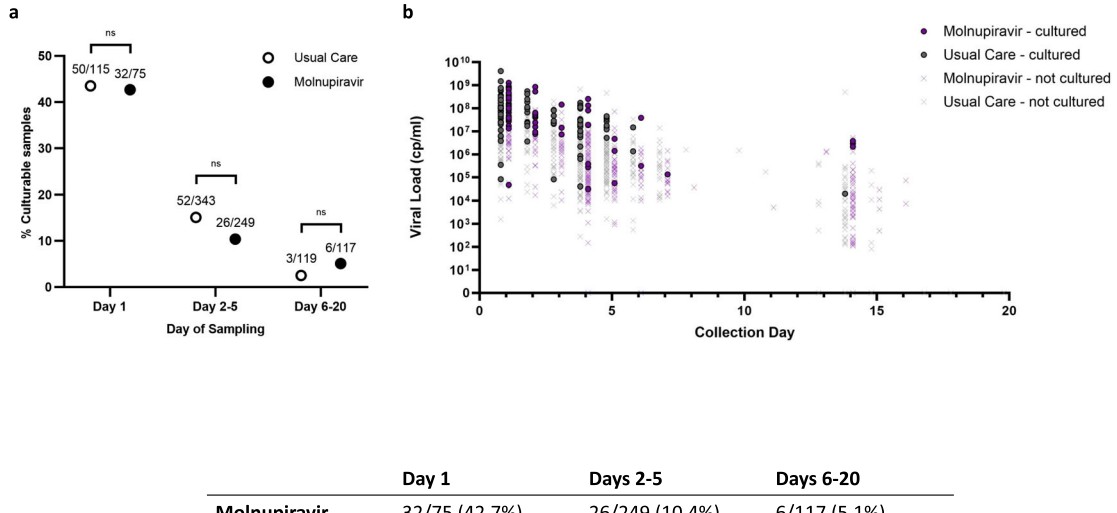

| | Day 1 | Days 2-5 | Days 6-20 |
|---|---|---|---|
| Molnupiravir | 32/75 (42.7%) | 26/249 (10.4%) | 6/117 (5.1%) |
| Usual Care | 50/115 (43.5% | 52/343 (15.2%) | 3/119 (2.5%) |

**Fig. 7 | Culture rate inferred from microscopy observation of cytopathogenic effects combined with a positive result by lateral flow immunochromatography of culture media, from nasopharyngeal swabs inoculated on to Calu-3 cells and incubated for 7 days. a** Overall culture rate split by baseline, on treatment and post treatment for each group compared with a two-sided Fisher's Exact Test. **b** Distribution of culturable samples by day of sampling versus viral load. Overall totals were 169 culture positive samples out of 1018 screened. No significant associations between treatment group or any of the tested covariates were found with probability of culture positivity in samples taken after Day 6. ns: *P > 0.05*. Source data are provided as a Source Data file.

molnupiravir is mutated and can be cultured up to 9 days post-cessation of treatment.

Our viral dynamic results conflict with those of the Phase II study[2], in which mean viral loads were lower and the proportion with <LLOQ viral loads higher in the molnupiravir-treated individuals one month later. However, in the Phase III trial viral load reductions were less marked during treatment, with less than 1 log10 copies/mL difference in treated or untreated by Day 5 and did not persist with post-treatment follow-up[1]. Our baseline viral loads were around 1 log10 higher than previous studies[2] and since our estimate of viral decline rate, δ, was similar to previous studies[14], our higher viral loads are a consequence of higher baseline. The causes of this include PANORAMIC participants being older with shorter time since symptom onset than previous studies. The incubation period of the omicron variant is around a day shorter than previous variants[15] which may have contributed to higher viral loads. In the treatment of infectious disease, it is rare for an effective therapeutic agent to be stopped before pathogen clearance. A 5-day course is too short to clear the virus and may explain the lack of clinical benefit in the main trial.

Modelling studies in SARS-CoV-2[16,17] and other acute viral infections[18] showed antivirals are most effective early in disease, and we were able to recruit participants within 2.5 days of symptom onset on average. Finding that male sex and increasing age associate with higher viral load correlates with these groups being most at risk of severe infection[19] and is in line with a previous study[20]. Lower spike antibody correlating with higher viral loads underlines the evidence for spike antibody being a correlate of protection and control[21]. The slower viral elimination rate post-treatment seen with molnupiravir maybe a result of greater numbers of target cells remaining to support ongoing infection[16,17], but lower spike antibody response due to initial viral suppression and/or viral genetic changes may contribute.

Anti-SARS-CoV-2 spike antibody titres were significantly lower with molnupiravir by Day 14 (Fig. 3d), even after controlling for other covariates (Fig. 3e). Early rapid reduction in viral load with molnupiravir treatment may have caused a smaller rise in antibody levels due to lower antigenic stimulation (Fig. 3f). Whilst we were unable to culture sufficient virus to undertake antibody neutralisation assays,

the magnitude of difference in antibody titre between the arms by Day 14 (2200 U/mL lower in the molnupiravir arm) when antibody response usually peaks, is likely to be clinically significant[22]. With populations becoming increasingly reliant on natural infection to boost immunity, antiviral blunting of antibody response is a concern[23].

Molnupiravir acts through transitional nucleotide mutagenesis. Where the virus is not driven to extinction, mutated but viable viruses persist[2,24]. Recently SARS-CoV-2 genome sequences potentially displaying the mutational hallmarks of molnupiravir treatment have been identified among opportunistically sampled sequences in public databases. The enrichment for these mutated sequences both geographically, in countries with higher molnupiravir use, and temporally, after molnupiravir received its first emergency use license, underlines concern that molnupiravir mutated viruses may be capable of transmission[25].

The ability of mutated viruses to persist fits with theoretical and experimental research in evolutionary biology suggesting that reductions in population fitness paradoxically confers increased potential for gaining beneficial variants[26]. It is possible that molnupiravir-induced mutagenesis leads to increased non-synonymous variants with improved viral fitness. The independent occurrence of nine consensus-level mutations in more than one molnupiravir-treated participant with virus detectable on Day 14 might indicate an element of selection. P104S (ORF3a) and S1952L (ORF1a), at a prevalence of 0.07% and 0.03% respectively, are rare in the UK population, based on the GISAID database of opportunistically collected samples, and were also found at baseline in some samples. The significance of these together with others in ORF1a and A99T (ORF3a) which are also frequently observed[12] is unclear.

The only repeatedly occurring substitution (F490S present in 1.52% of the UK population) in the RBD has been shown to affect the binding affinity to the ACE2 receptor. F490S has also appeared naturally in two variants of concern, the lambda lineage in 2021 and the currently circulating XBB 1.5 and related lineages[27–31]. The rise to consensus following treatment, of F490S independently in two participants with virus persisting at Day 14, may be due to selection.

One consensus mutation and another below consensus that potentially could cause molnupiravir resistance based on data from MERS-CoV[32] were found. V234I was present at consensus in one Day 14 virus but not in an earlier sample. It was not present at baseline in this population and found at very low frequency (1–5%) in three molnupiravir-treated participants at Day 5. V560L was present below consensus at Day 14 and not found in other participants. The independent emergence of the A716V mutation in the RdRp catalytic binding site on Day 14 in more than 1 patient is notable. This mutation was only found in molnupiravir-treated samples, and only rising to consensus level in two participants with virus persisting at Day 14. A716V has a prevalence in the UK of 0.0013% (GISAID database). Although the low levels seen during treatment make it unlikely to have conferred high level resistance to molnupiravir, it is possible that A716V, having been selected by molnupiravir, promotes greater survival fitness. Marker transfer studies to examine resistance for all three putative resistance substitutions are now needed.

Previous studies recovered viable virus from placebo-treated groups but not after treatment with molnupiravir[2] whereas we did recover viable virus after molnupiravir treatment. The high transition to transversion mutation ratio we observed is in line with observational data[10]. These observational data also had long branch lengths with evidence of excess transition (G to A and C to T) mutations. We observed 224 consensus genome-wide amino acid substitutions in viruses recovered from molnupiravir-treated participants on Day 14 (Supplementary Table 6). Only one substitution in the spike region was also reported in the opportunistically collected samples[25]. Other mutations below consensus were also found in both datasets. There were six mutations in the Usual Care group. The methods developed by Sanderson et al.[10] identified persistent molnupiravir-treated viruses in our study with high specificity, thus confirming their utility as public health tools for monitoring circulation of these viruses. Our data support the hypothesis that molnupiravir-mutated viruses which survive treatment can remain viable, thus posing a risk for transmission of new variants.

Sequencing identified four participants whose viral sequences showed little or no evidence of drug exposure, meaning they may not have taken the drug despite reporting having done so. For a further nine participants, who, despite evidence for molnupiravir-associated mutations at Day 5, had lost these by Day 14, it is possible that wild type virus, sequestered in compartments protected from drug treatment, outcompeted mutated virus when molnupiravir was terminated. Incomplete drug penetration into inflamed lungs has been proposed to explain similar findings in SARS-CoV-2 and influenza[33]. In such situations, combination treatments may be beneficial[34].

Systemic inflammation, especially CRP > 40 mg/L[35] and elevated interleukin-6[19] is a hallmark of severe disease with COVID-19. Baseline disease severity as measured by CRP demonstrated a positive correlation with baseline viral load. Importantly, molnupiravir appeared to have no impact on CRP. This contrasts with findings from the MOVe-OUT study where post-baseline CRP levels were lower in the molnupiravir treated group[36]. Our dry blood spot CRP assay is exploratory but its positive correlation with baseline viral load and reduction over time suggest it reflects inflammation. Therefore, differences in baseline immune status (SARS-CoV-2 naïve in MOVe-OUT but vaccinated in PANORAMIC), and possibly lower overall degree of systemic inflammation at baseline in PANORAMIC may be more relevant.

Our study has some limitations. Since recruitment was from across the UK, participants were required to self-sample and return these by post. Of the participants who received kits we had an excellent rate of sample returns (Table 1). High sample quality likely resulted from participants being provided detailed written and video instructions. Treatment allocation was not blinded but agreement to take part was before randomisation so higher numbers in Usual Care likely appeared by chance. The two arms were matched for demographics

and baseline variables except for more female participants in Usual Care. Reassuringly the proportion returning samples (90% versus 88%) was not significantly different between the groups (p = 0.2706). Multivariable models corrected for sex, viral load and spike antibodies were similar between groups at baseline, only becoming significantly different with or without molnupiravir. This unbalance is therefore unlikely to have biased the results. Our study was not designed or powered to detect rates of viral mutation frequency, rather to ensure enough viruses were sequenced to detect common mutations and therefore generate hypotheses on consequences of mutagenesis. We fell short of the overall target of 600 participants before the molnupiravir arm closed due to the time lag between recruitment and sample return, but did achieve the minimum desired 30 per arm in the intensively samples group. We were unable to confirm escape of mutated viruses from the effects of host antibodies through viral neutralisation as culture was unsuccessful for these samples.

In summary, following a 5-day molnupiravir course, whilst viral load decreased on treatment it was not reduced to <LLOQ levels before the drug was stopped, thus violating a fundamental principle of antimicrobial chemotherapy. Post-treatment, higher viral loads occurred in the molnupiravir group and some of these viruses were culturable indicating potential for transmission. Molnupiravir-exposed viruses were highly mutated, and the increased numbers of functionally relevant mutations including those known to confer evasion of neutralising antibody, potentially conferring selective advantage, is of concern. Furthermore, spike antibody response was blunted. Our study therefore indicates that 5 day of molnupiravir could drive the emergence of SARS-CoV-2 variants capable of increased virulence or transmission and potentially leaves treated patients more susceptible to re-infection. These findings imply a need for a longer course of molnupiravir to be tested in a clinical trial and should inform decisions on whether to use the current 5-day course of molnupiravir until the results of such a trial are available.

## Methods
### Sample Collection
To be eligible to join the PANORAMIC study (ISRCTN30448031), participants had to be symptomatic within the past 5 days, have a confirmed COVID-19 infection within the past 7 days (either a positive PCR or rapid antigen SARS-CoV-2 test), and be either aged 50 years and over or at least aged 18 years with relevant comorbidities. Participants provided written informed consent to take part in the trial and a subset were invited to take part in the virology sub-study for which further written informed consent was obtained. The UK Medicines and Healthcare products Regulatory Agency and the South Central-Berkshire Research Ethics Committee of the Health Research Authority approved the trial (Ethics approval reference: 21/SC/0393). Participants were not compensated financially or otherwise for taking part. Details of the main study design and statistical analysis plan have been described previously[8].

The virology study's pre-specified primary outcome was viral load at Day 7, and has been reported previously[8]. The pre-specified secondary outcomes reported here were: change in viral load over 14 days post-treatment, change in spike antibody concentrations over 14 days post-treatment, and to determine whether genetic mutations in the virus are more frequent in patients taking antiviral treatment compared with Usual Care.

Participants in the virology sub-study were provided with Respiratory Virus Swab Collection kits, which consisted of a Sigma Virocult 1 ml tube containing Liquid Virocult medium and a Standard Tip Swab that complied with the European In-Vitro Diagnostic Devices Directive. The kits also included detailed instructions approved for self-sampling of nasal and pharyngeal swabs, as well as dried blood spot sampling kits. A weblink to a video tutorial was provided to aid sample collection (https://youtu.be/klB-ckiQGz8).

The goal was to recruit 300 participants into the trial intervention arm (molnupiravir) and the Usual Care comparator arm. Based on a previously published viral dynamic model[17] it was estimated that 30 participants in each arm could detect a proportional difference of 0.36 to 0.8 in below-detection-limit viral loads based on a two-sample test for proportions with a Type I error probability set to 0.025 and power set to 0.9. Hence this would give sufficient power to detect viral load differences for a potent antiviral and was the sample size for the Intensive sampling group. A further 270 participants in each arm had less-intensive viral load monitoring with the aim to detect smaller viral load differences and, since genetic mutations may not happen under antiviral treatment and if they do, their frequency was unpredictable, a total of 300 participants would provide a 95% probability of seeing at least one example of a mutation occurring in 1% or more of participants.

In the intensive sampling cohort, the participants were asked to collect nasopharyngeal swabs daily from Days 1 to 7 and on Day 14. In the non-intensive sampling cohort, participants collected nasopharyngeal swabs on Days 1, 5, and 14. Participants in both arms provided the baseline Day 1 sample upon receipt of the sampling kit, which for those randomised to receive molnupiravir arrived on the same day as the drug. The molnupiravir group were asked to provide their first sample before the first dose of molnupiravir. All samples were sent to the Great Ormond Street Hospital laboratory via Royal Mail priority boxes for prompt delivery. RNA samples positive for SARS-CoV-2 were then transferred to UCL Genomics for sequencing. All participants in the virology sub-study were also asked to provide three finger-prick dried blood spot samples on Days 1, 5, and 14.

## Sample processing
Upon arrival at the Great Ormond Street Hospital laboratory, the swabs in VTM were agitated on a vortex mixer, and 250 μl of the sample fluid was transferred to a sterile Sarstedt tube containing lysis buffer for RNA extraction using the Microlab STAR platform. To control for extraction failures, PDV (Phocine Distemper Virus) was spiked into every sample during the extraction process. Negative sample extraction controls were included in each batch of extractions. After extraction, the samples were tested for SARS-CoV-2 and PDV using multiplex targeted one-step real-time RT-PCR on a Quantstudio 5 instrument. Each PCR run included a SARS-CoV-2 positive template control and a no-template control. Extraction and PCR control results were recorded and monitored for consistency.

## Viral load
Viral loads were determined using a standard curve generated from a ten-fold dilution series of SARS-CoV-2 RNA with a known quantity. This curve was imported into the PCR analysis to calculate the viral load in copies/mL for each positive sample. The lower limit of quantification (LLOQ) was set at 109 copies/mL corresponding to a Ct value of 40. RNA samples positive for SARS-CoV-2 with a Ct value below 38 were subsequently sent to UCL Genomics for sequencing.

## Sample storage and culture
To allow for subsequent culture, a portion of the original sample was aliquoted into a tube containing Bovine Albumin Fraction V (7.5%) from Gibco, resulting in a final bovine serum albumin concentration of 0.4%. Samples mixed with BSA were stored at 4 °C and then moved to storage −80 °C within 24 h. The remaining sample was stored separately at −80 °C to allow for potential repeat RNA extraction.

## Sequencing
Libraries were prepared on an Agilent Bravo workstation using the CoronaHiT protoco[37] with ARTIC v4.1 amplicon primers. Sequencing was performed aiming for a target depth of 5000X per genome on an Illumina sequencer using 2 x ≥ 75-bp paired-end reads. All runs

included both positive and negative controls to detect contamination at each step of the protocol.

## Antibody assays and CRP assays
Dried blood spot cards were processed as previously described[38]. Anti-SARS-CoV-2 spike antibody titres were measured in elutes using a commercial immunoassay (Elecsys Anti-SARS-CoV-2 S, Roche) with a validated 10-fold dilution factor correction applied. CRP concentration was determined by turbidimetry (Roche/Hitachi Cobas c) with no correction factor applied. Values are predicted to be approximately 10-fold lower than serum concentrations.

## Statistical analysis, modelling and data visualisation
Statistical analyses and data visualisation were undertaken in R (version 4.3). Descriptive statistics and plots of baseline characteristics (baseline viral load and baseline spike antibody versus each other and covariates) were created along with plots of viral load, spike antibody and CRP evolution in time. Pairwise comparisons of the median of viral load were compared at each time point using a Mann-Whitney test with samples below the limit of quantification being replaced with the LLOQ/2.

Log10 viral load and spike antibody concentrations were each modelled with time using a linear mixed effects model in the R package nlmixr2. We originally planned to use a nonlinear viral dynamic model, but subsequent work completed after the statistical analysis plan was finalised showed viral decline could equally well be described using a linear model[39]. Maximum likelihood estimation was used to fit all observations including those that were censored, with the probability of LLOQ observations being below the limit of quantification estimated ("M3 method")[40]. A stepwise approach to covariate analysis was taken whereby variables (sex, time since symptom onset, baseline spike antibody, age) were tested sequentially on baseline viral load and slope. Dichotomous covariates were tested with a proportional effect whereas for continuous variables an allometric exponent was estimated as follows: $p_i = p \times (c_i/c_t)^\beta$, where $p_i$ is the individual parameter, $p$ the population parameter, $c_i$ the individual covariate value, $c_t$ the median covariate value in the population and $\beta$ the estimated covariate effect. The significance of additional parameters was evaluated with a likelihood ratio test, the difference in −2 log-likelihood in the models being asymptotically Chi-square distributed with one degree of freedom. Covariates were included if the likelihood ratio test indicated a significant improvement in fit at the level of $p < 0.01$. A similar approach was taken to modelling the increase in spike antibody over time.

The molnupiravir drug effect in the viral dynamic model was estimated with a time-varying covariate on viral decline slope[39]. A further effect of slower viral decline following the end of treatment was also estimated. The final multivariable model was used to simulate different durations of treatment. A simulated dataset of 1000 participants with demographics sampled from the virology sub-study demographic database was created. Each simulated participant was assigned model parameters based on their demographics and sampling from the parameter distribution. The drug effect was increased in daily increments with successive simulations and the probability of virus being below the LLOQ reported for each treatment duration. This was repeated 1000 times to generate 95% prediction intervals.

The viral and antibody models were also used to extract individual parameter estimates for each participant. These were then used to calculate the area under the curve (AUC) from Day 0-14 using the trapezoidal rule on Log10 transformed individual model predictions of viral load and compared with individual predicted spike antibody doubling times.

## Sequence analysis
The raw fastq reads were adapted, trimmed and low-quality reads removed using the fastp algorithm. The reads were aligned against the

Wuhan-Hu-1 reference genome (NC_045512.2, EPI_ISL_402125) reference sequence and then the amplicon primers regions were trimmed using the location provided in a bed file. Consensus sequences were then called at a minimum of 30X coverage. The entire processing of raw reads to consensus was carried out using nf-core/viralrecon pipeline[41] (https://nf-co.re/viralrecon/2.6.0). Only samples producing genomes with at least 90% genome coverage at 10X sequencing depth were kept for further analysis. Variant calling was done using the iVAR algorithm[42]. Mapped data was also used to calculate the Shannon entropy measurements that are then aggregated into a single score per sample, using the DiversiTools approach[43].

### Mutation analysis
Post-baseline mutations were calculated by subtracting baseline mutations (nucleotide) in the additional time points per each participant, moreover lineage specific mutations were removed from the dataset. Two databases, Pokay[10] and SARS2-ResistanceDB[11], were used to investigate mutations potentially associated with immune system escape or drug resistance to molnupiravir, respectively.

A multivariable linear regression was used to compare the maximum post-baseline mutation load, transitions and transversions in each participant. The covariates in the model were: treatment group, age, sex, log10 baseline viral load, log10 baseline spike antibody and the comorbidities: lung, heart, kidney, liver, neurological and immune disease, being a transplant recipient, obesity and hypertension.

### Phylogenetic analysis
A multiple sequence alignment of the consensus sequences obtained for each sample was performed using the MAFFT (Multiple Alignment using Fast Fourier Transform) algorithm[44]. A phylogenetic tree was constructed from the multiple sequence alignment using the IQ-TREE[45] algorithm using automatic model selection and rapid bootstrap. Samples were also compared with the global SARS-CoV-2 phylogeny using the UShER algorithm[46].

### Viral culture
Human epithelial airway Calu-3 cells (obtained from the Francis Crick) were grown to 50–80% confluency on 24 well plates (Corning) in maintenance medium consisting of OptiMEM (Gibco) supplemented with 5% FBS (Gibco), 1% penicillin/streptomycin (Thermo Fisher) and 250 ng/ml amphotericin B (Invitrogen). They were then inoculated with 100-200 μL of residual swab transport media for 1–2 h at 37 °C and 5% $CO_2$. The inoculum was then replaced by maintenance medium and cells were maintained at 37 °C and 5% $CO_2$ for 7 days. At interval (Day 4–5) and endpoint (Day 7) of infection all wells were screened for cell confluence and observable cytopathic effects (CPE) using an inverted phase contrast microscope (Olympus CK40). The supernatants from wells with suspected CPE were further tested for presence of SARS-CoV-2 nucleocapsid antigen by lateral flow immunochromatography (ACON Biotech). Supernatants were collected and stored at −70 °C for further culture or analysis when CPE or cell death exceeded 50% of the well, or at the Day 7 endpoint of infection.

A multivariable generalised linear model was used to investigate covariates associated with the probability of samples being culture positive in post-treatment samples (Day 6 and above). The covariates in the model were: treatment group, age, sex, log10 baseline viral load and spike antibody, and the comorbidities: lung, heart, kidney, liver, neurological and immune disease, obesity and hypertension.

### PCR of cultured viral supernatants
Collected frozen supernatants, as described above, were thawed and inactivated by the addition of RNA/DNA shield (Zymo Research) prior to RNA extraction using the QIAamp Viral RNA Mini Kit (Qiagen). 100 ng of extracted RNA from each sample was added to TaqMan™

Fast Virus 1-Step Master Mix and TaqMan™ gene expression assay Vi07918637_s1 (Thermo Fisher Scientific) directed towards the SARS-CoV-2 N gene, according to the manufacturer's instruction. The qRT-PCR was run on the StepOnePlus™ Real-Time PCR System for 5 min at 50 °C followed by 20 s at 95 °C and then 40 cycles of 3 s at 95 °C followed by 30 s at 60 °C. StepOne™ Software was used to generate the Ct values shown.

## Data availability
Whole genome sequencing data generated in this study have been uploaded to the CLIMB repository (https://docs.covid19.climb.ac.uk/accessing-data.html) and is available to GISAID. The COG-UK accession numbers are listed in Supplementary Table 7. Additionally the sequence data have been uploaded to the NIH BioProject no PRJNA1066240. De-identified source data has been made available in the source data file except where de-identification was not feasible. Selected, further de-identified individual participant data for outcome measures will be available on request, accompanied by a protocol outlining hypotheses and proposed analytic methods, by contacting the corresponding author (panoramic@phc.ox.ac.uk): requests will be considered by a Departmental Committee. A data sharing agreement will be required. Source data are provided with this paper.

## Code availability
The viral and antibody dynamic model code is available in the Zendo repository under accession code https://doi.org/10.5281/zenodo.10375295

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

## Acknowledgements

The authors would like to thank all participants in the PANORAMIC virology sub-study. This study was funded by the UK NIHR (NIHR135366) to C.B. and its design and analysis methods supported by a grant from the UK MRC (MR/X004724/1) to J.F.S. The funders had no role in designing the study. The PANORAMIC Trial Collaborative Group is also acknowledged and listed in the Supplementary Material. This paper is dedicated to our co-author Oghenekome (Kome) Gbinigie who sadly passed away in January 2024.

## Author contributions

J.F.S, D.M.L. and J.B. conceived and designed the virology sub-study. J.D., O.G., O.V.H., M.L., N.F., B.J., D.B.R., N.M.R., N.P.B.T., N.H., P.E., M.A., K.H., F.D.R.H., G.H., J.S.N.v.T., S.K., P.L. and C.B., undertook and oversaw patient inclusion and clinical trial conduct. M.W., S.E., C.M., A.I.J., C.W., S.R. L.M.M., F.A., B.E., Z.M.A., R.W., A.S., C.M.S., M.O., E.K., and A.R. were responsible for the execution and quality control of the laboratory work and data processing. J.F.S., L.B., J.A.G-A., S.E., A.A., T.S., L.M.Y., D.M.L.,

and J.B. analyzed the data. J.F.S., L.B., S.E., D.M.L. and J.B. wrote the paper. All authors significantly contributed to interpreting the results, critically revised the manuscript for important intellectual content and approved the final manuscript.

## Competing interests

J.F.S. has participated on a data safety monitoring board for GlaxoSmithKline (Sotrovimab) with fees paid to his institution. JSN-V-T was seconded to the Department of Health and Social Care, England (DHSC) from October 2017–March 2022. The views and opinions expressed in this paper are not necessarily those of DHSC or any of its arms-length bodies. JSN-V-T performed one-off paid consultancy for Merck Sharp and Dohme in June 2023, unrelated to the subject of the manuscript. K.H. was a member of the Health Technology Assessment General Committee and Funding Strategy Group until November 2022, and Research Professors Funding Committee at the UK National Institute for Health and Care Research (NIHR), received a grant from AstraZeneca (paid to their institution) to support a trial of Evusheld for the prevention of COVID-19 in high-risk individuals (RAPID-Protection), and was an independent member of the independent data monitoring committee for the OCTAVE-DUO trial of vaccines in individuals at high risk of COVID-19. D.M.L. has received grants or contracts from LifeArc, the UK Medical Research Council, Bristol Myers Squibb, GlaxoSmithKline, the British Society for Antimicrobial Chemotherapy, and Blood Cancer UK, personal fees or honoraria from Biotest UK, Gilead, and Merck, consulting fees from GlaxoSmithKline (paid to their institution), and conference support from Octapharma. DBR has received consulting fees from OMASS Therapeutics, GSK, and Sosei-Heptares and has a leadership and fiduciary role in the Heal-COVID trial TMG. M.L. is a member of the data monitoring and ethics committee of RAPIS-TEST (NIHR efficacy and mechanism evaluation). S.K. reports grants from GlaxoSmithKline, ViiV, Ridgeback Biotherapeutics, Vir, Merck, the UK Medical Research Council, and the Wellcome Trust (all paid to his institution), speaker's honoraria from ViiV, and donations of drugs for clinical studies from ViiV Healthcare, Toyama, and GlaxoSmithKline. M.A. has received grants from the Blood and Transplant Research Unit, Janssen, Pfizer, Prenetics, Dunhill Medical Trust, the BMA Trust (Kathleen Harper Fund), and Antibiotic Research UK (all of which were paid to their institution), and consultancy fees from Prenetics and OxDx. M.A. reports a planned patent for Ramanomics, has participated on data safety monitoring boards or advisory boards for Prenetics, and has an unpaid leadership or fiduciary role in the E3 Initiative. NPBT has received payment for participation on an advisory board from MSD (before any knowledge or planning of this trial). O.v.H. has received consulting fees from MindGap (fees paid to Oxford University lnnovation), has participated on data safety monitoring boards or advisory boards for the CHICO trial, and has an unpaid leadership or fiduciary role in the British Society of Anti-microbial Chemotherapy. J.B. has received consulting fees from GlaxoSmithKline (paid to her institution). All other authors declare no competing interests.

## Additional information

Joseph F. Standing [1,2] ✉, Laura Buggiotti [1], Jose Afonso Guerra-Assuncao[1], Maximillian Woodall[1], Samuel Ellis[1], Akosua A. Agyeman[1], Charles Miller[2], Mercy Okechukwu[1], Emily Kirkpatrick[1], Amy I. Jacobs[1], Charlotte A. Williams[3], Sunando Roy[3], Luz M. Martin-Bernal[3], Rachel Williams[3], Claire M. Smith [1], Theo Sanderson [4], Fiona B. Ashford [5], Beena Emmanuel[5], Zaheer M. Afzal[5], Adrian Shields [5], Alex G. Richter[5], Jienchi Dorward [6,7], Oghenekome Gbinigie[6,26], Oliver Van Hecke[6], Mark Lown[8], Nick Francis[8], Bhautesh Jani[9], Duncan B. Richards [10], Najib M. Rahman [11], Ly-Mee Yu [6], Nicholas P. B. Thomas[12], Nigel D. Hart [13], Philip Evans [14,15], Monique Andersson [16], Gail Hayward[6], Kerenza Hood [17], Jonathan S. Nguyen-Van-Tam [18], Paul Little[8], F. D. Richard Hobbs[6], Saye Khoo [19], Christopher Butler [6], David M. Lowe [20,21,25], Judith Breuer [1,2,25] and PANORAMIC Virology Group*

[1]Infection, Immunity and Inflammation, Great Ormond Street Institute of Child Health, University College London, London, UK. [2]Great Ormond Street Hospital for Children NHS Trust, London, UK. [3]Genetics and Genomic Medicine Department, Great Ormond Street Institute of Child Health, University College London, London, UK. [4]Francis Crick Institute, London, UK. [5]Clinical Immunology Service, Institute of Immunology and Immunotherapy, University of Birmingham, Birmingham, UK. [6]Nuffield Department of Primary Care Health Sciences, University of Oxford, Oxford, UK. [7]Centre for the AIDS Programme of Research in South Africa (CAPRISA), University of KwaZulu–Natal, Durban, South Africa. [8]Primary Care Research Centre, University of Southampton, Southampton, UK. [9]School of Health and Wellbeing, University of Glasgow, Glasgow, UK. [10]Nuffield Department of Orthopaedics, Rheumatology and Musculoskeletal Sciences, University of Oxford, Oxford, UK. [11]Respiratory Trials Unit and Oxford NIHR Biomedical Research Centre, Nuffield Department of Medicine, University of

Oxford, Oxford, UK. [12]Windrush Medical Practice, Witney, UK. [13]School of Medicine, Dentistry and Biomedical Sciences. Queen's University Belfast, Belfast, UK. [14]APEx (Exeter Collaboration for Academic Primary Care), University of Exeter Medical School, Exeter, UK. [15]National Institute of Health and Care Research, Clinical Research Network, University of Leeds, Leeds, UK. [16]Radcliffe Department of Medicine, University of Oxford, Oxford, UK. [17]Centre for Trials Research, Cardiff University, Wales, UK. [18]Lifespan and Population Health, University of Nottingham School of Medicine, Nottingham, UK. [19]Department of Pharmacology, University of Liverpool and Liverpool University Hospitals NHS Foundation Trust, Liverpool, UK. [20]Department of Clinical Immunology, Royal Free London NHS Foundation Trust, London, UK. [21]Institute of Immunity and Transplantation, University College London, London, UK. [25]These authors contributed equally: David M. Lowe, Judith Breuer. [26]Deceased: Oghenekome Gbinigie. *A list of authors and their affiliations appears at the end of the paper. ✉e-mail: j.standing@ucl.ac.uk

## PANORAMIC Virology Group

Julie Allen[22], Nadua Bayzid[3], Julianne Brown[2], Doug Burns[1], Elizabeth Hadley[23], Jim Hatcher[2], Tim McHugh[24], Chris Thalasselis[1], Mia Tomlinson[1] & Francis Yongblah[2]

[22]IQVIA Global Clinical Trials, London, UK. [23]Faculty of Medicine, Imperial College, London, UK. [24]Centre for Clinical Microbiology, University College London, London, UK.

