## [Peer Review File · Nature Communications]

REVIEWER COMMENTS

Reviewer #1 (Remarks to the Author):

NCOMMS-23-35977-T

Title: Early treatment with five days of molnupiravir reduces SARS-CoV-2 viral load but lowers spike antibody response and mutated virus persists

Overview: This is well-written paper describing the use of molnupiravir in the PANORAMIC trial. It was very interesting that Day 14 viral loads, slight culture-ability and antibody levels were higher in molnupiravir-treated participants than Usual Care. It was also good to see consistency with previous studies of increased mutations in virus from the MOL treated arm.

Major comments:

- Need to provide data for specific variants evaluated during the study.
- Need to describe which participant factors were associated with Day 14 viral loads, slight culture ability and antibody levels that were higher in molnupiravir-treated participants than usual care
- Need to describe which participant factors were associated with drug associated mutagenesis
- As stated , perhaps MOL should be given longer but needs to be rigorously tested in a clinical trial and not just in a modeling study. The current study does not assess this so such broad claims cannot be made. Please temper this language

Minor comments

- Discussion could be shortened
- In the discussion of antibody blunting should cite this paper: Neutralizing Antibody Responses After Severe Acute Respiratory Syndrome Coronavirus 2 BA.2 and BA.2.12.1 Infection Do Not Neutralize BA.4 and BA.5 and Can Be Blunted by Nirmatrelvir/Ritonavir Treatment - PubMed (nih.gov)
- Similarly, it would be good to look at neutralization of antibodies after treatment, like the Carlin et al paper. If not, then at least note this in the limitations of the paper.

Reviewer #2 (Remarks to the Author):

This is an interesting and very important study concerning the inadequacy of a five-day course of treatment of outpatient COVID-19 with molnupiravir, both from a treatment perspective and from the perspective of risk of transmission of resistant virus. I have the following comments followed by some specific details that it would be useful to address:

1. The difference between arms in number of participants agreeing to provide additional less intensive sampling for the virology sub-study seems surprisingly large for a randomized trial (315 for Usual Care versus 248 for Molnupiravir). If agreement to participate was obtained prior to randomization then it would seem to be an unlucky chance imbalance but if agreement was obtained after randomization then it raises the possibility of bias in comparisons between arms beyond the factors that you adjusted for. Related to this:

a. Please clarify whether agreement was obtained before or after randomization and, if after, was it before or after a participant was made aware of their randomized assignment? (Extended data fig 1 seems to imply that it was after randomization so suggest reworking that figure if agreement was obtained before randomization).

b. If agreement was after randomization, then the limitations section should recognize the possibility of confounding due to factors, measured or unmeasured, beyond those that you adjusted for.

c. A suggestion is to give the proportions who returned samples by arm in lines 140 to 143, to establish that these are reasonably similar and unlikely a source of bias.

2. Figure 2a is key for major conclusions but has the limitation that large numbers of measurements below the LLOQ are being imputed using the value LLOQ/2. A better presentation would be to plot the medians (alternatively, if assumptions are satisfied, use regression for censored data to estimate the mean at each time point). Please also clarify why the CIs at Day 4 are narrower than those at Day 5 despite what should be a much larger sample size at Day 5 (or did “less intensive” participants often test a day early, in which case might they also have stopped taking treatment a day early? – what were the specific instructions for Day 5 testing?).

3. The primary publication for the trial mentions participant-reported adherence information. It would be good to include a summary of that for the substudy population to establish whether or not participant-reported adherence was very high. The interpretation of results would need to be more cautious if participant-reported adherence was not very high.

4. A simulation study is used to estimate the probability of viral load being <109 copies/mL if molnupiravir was continued after 5 days. Please state in lines 173-4 the model being used to simulate this (presumably it is an extrapolation of the model over Days 1 to 5?). It should be made clear here and in the Abstract and Discussion that this is an extrapolation which cannot be validated based on data from this study. Please also detail to line 585-591 to clarify what was done.

5. Line 326-327: Please provide some additional underpinning to this conclusion. It suggests that most of the hospitalizations/deaths were after the end of treatment so it would be good to show cumulative incidence curves for hospitalization/death to confirm this.

6. Discussion: Suggest contrasting your findings concerning predictors of baseline viral load and longitudinal changes with that of Moser et al (Open Forum Infect Dis, 2022), particularly regarding findings for sex differences.

Specific details:

Throughout: Some assays detect virus below the LLOQ. So please clarify whether “undetectable” means target not detected or whether it means below the LLOQ. If the latter, suggest using, e.g., “<LLOQ” rather than “undetectable”. Similarly, “viral clearance” in lines 173/4 should be changed to “<LLOQ” or “<109 copies/mL” as used in the legend of Figure 2).

Abstract: Please add that these are non-hospitalized adult participants.

Line 182 and Figure 2f: Based on the units for the AUC (copies*days/mL), it seems that the AUC is being calculated using the unlogged viral loads and hence is likely driven by the Day 1 level and so initial virus exposure---perhaps spike antibody time is similarly correlated with Day 1 viral load? It would be good to check and clarify this. Also clarify whether AUC is model-based, or based on actual measurements.

Line 205: Is the substantial difference in rate of successful sequencing between arms due to the reasonably modest difference in viral load shown in Figure 2a? It would be good to explain the difference to reassure the reader that other factors are not causing the difference.

Line 213: why 45 and not 46 as in line 205?

Lines 309-10 and 171-172: This is a key finding, presumably based on 44% for Usual Care and 52% for Molnupiravir being above LLOQ? It would be good to be explicit about this and that there is a significant increase at Day 14.

Line 317: Add “copies/mL” before “difference”.

Line 493: It seems that this sentence is about the intensive sampling. If so, it would be good to clarify that.

Lines 527 to 533: What was the upper limit of quantification for this assay and how were results above this handled in the modeling? (or was there retesting of diluted samples if an initial result was above an upper limit?).

Line 563: See above comment about avoiding using imputed values in the analysis. Also what were the categorical variables being tested?

Line 567-9: Please confirm that the linear model provided adequate fit in this specific study.

Line 570: “probability” might be better than “likelihood”.

Lines 573-4: Please clarify what you mean by “on baseline viral load and slope”.

Line 576: i should be subscript in “ π ”.

Line 611: Please clarify what you mean by “were adjusted for viral load”.

Figure 1 legend: Note your criterion for selecting when to show p-values.

Extended data tables 2 and 3: (a) Add units for all parameters (and give correct units for delta). (b) "Additive error variance" might be clearer, with units stated also. (c) Add an algebraic definition of the model being described (this likely will help some readers understand the parametrization).

Extended data figure 1: (a) You have $n=6131$ splitting into $n=3093+3034=6127$. Please correct/clarify (line 134 also mentions 6127 so perhaps that is incorrect also?). (b) Add an additional row of boxes at the bottom showing the number included in the analysis (this will also clarify whether or not the exclusions are included in the count of participants who returned samples).

Extended data figure 6: The legend mentions $n=558$ and $n=425$ "participants" but these are larger than the study population. Please clarify. If it is counting multiple samples per participant, then how did you address this in statistical analysis?

Supplementary Figure 1a: I'm unclear what are the DV values shown when they are block dots and so below LLOQ?

Reviewer #3 (Remarks to the Author):

Standing et al. performed detailed analysis of SARS-CoV-2 dynamics and evolution in molnupiravir-treated individuals, with several key findings: 5 days of molnupiravir reduces viral load but does not result in virus clearance; it appears to blunt the antibody response; and there are mutated but replication-competent viruses remaining at 2 weeks after treatment initiation. Overall, this is a really well-done and well-described study that adds valuable insight into the use of molnupiravir treatment for SARS-CoV-2. I have a few questions and comments:

1. Please provide more details about the multivariable model that was used to evaluate parameters associated with viral load decline (lines 166-171). The beta coefficients from extended data 2 should be discussed in the main text to help the reader understand the magnitude of change. In extended data 2, it also needs to be more clear which beta coefficients affected baseline viral load vs. slope; it took several reads of the legend and I'm still not sure I understand this. Suggest adding another column or horizontal breaks in the table to indicate which parameter of the model was affected by each variable.

Same comments for the analysis of spike antibody concentrations (lines 176-184).

2. Please describe how simulations of the viral dynamic model were performed (lines 172-174). I don't see this in the methods.

3. Using a minimum cutoff of 10X sequencing depth (line 188) does not square with analyzing SNPs down to 1% frequency (extended data figure 3), which would require over 100X depth. I am also not convinced that read depth and SNP detection aren't associated. It's not clear why the bins in Supplementary Figure 4 were set the way they are (<250, 250-500, 500-750), but a scatter plot with correlation analysis would be more convincing.

4. The statement that "emerging spike antibody mutations... were possibly associated with slower decline" (lines 239-241) should include a result of statistical analysis, because the slopes in Figure 5b don't appear different. Additionally, total number of mutations is an important confounder in this analysis.

5. The statement that "we did not find a major difference in the proportion with recoverable virus between groups" (line 296) should include results of a statistical analysis.

6. Please modify the discussion about the duration of treatment in other infectious diseases (lines 324-326, 462-464); while I agree with the authors that their study supports a longer treatment course for molnupiravir, we almost never perform tests of clearance before deciding to stop antimicrobial therapy for other infections and I doubt we would do so for SARS-CoV-2.

Reviewer #1 (Remarks to the Author):

Title: Early treatment with five days of molnupiravir reduces SARS-CoV-2 viral load but lowers spike antibody response and mutated virus persists

Overview: This is well-written paper describing the use of molnupiravir in the PANORAMIC trial. It was very interesting that Day 14 viral loads, slight culture-ability and antibody levels were higher in molnupiravir-treated participants than Usual Care. It was also good to see consistency with previous studies of increased mutations in virus from the MOL treated arm.

We thank the reviewer for this positive statement and provide point-by-point responses below.

Major comments:

- **Need to provide data for specific variants evaluated during the study.**

This information was provided in our original submission with a general statement in the main text and detailed breakdown in Supplementary Table 2:

Samples were collected between March and April 2022 with lineage analysis assigning over 70% to clade Omicron BA.2 (72.9% in molnupiravir-treated participants and 75.2% in Usual Care; Supplementary Table 2).

- **Need to describe which participant factors were associated with Day 14 viral loads, slight culture ability**

Since the original submission we have continued to attempt to culture samples. We have now attempted to culture 142 Day 14 samples. We also looked at any post-treatment sample (Day 6 onwards) and in this case 9/236 were positive. Fitting a multivariable generalised linear model revealed no factors significantly associated with culture positivity either at Day ≥ 6 or Day 14. The following text has been added:

Methods: A multivariable generalised linear model was used to investigate covariates associated with the probability of samples being culture positive in Day 14 samples. The covariates in the model were: treatment group, age, sex, log₁₀ baseline viral load and spike antibody and the comorbidities: lung, heart, kidney, liver, neurological and immune disease, obesity and hypertension.

Figure 6 legend: No significant associations between treatment group or any of the tested covariates were found with probability of culture positivity in samples taken after Day 6.

...and antibody levels that were higher in molnupiravir-treated participants than usual care

This point was addressed in our original submission. The antibody dynamic model, taking all antibody samples at all time points, was analysed with a multivariable mixed effects analysis. This analysis corrected for treatment effect. The only participant factors therefore associated with antibody levels were being male and not fully vaccinated. As per comments by Reviewers 2 and 3, we have now made interpretation of this model more explicit.

- **Need to describe which participant factors were associated with drug associated mutagenesis**

This is an important comment that we did not fully address. We have now undertaken a multivariable analysis on maximum post-baseline mutation load, transition/transversion ratio and spike escape mutations. Beyond molnupiravir treatment, which was significant for mutations and transition/transversion ratio but not spike escape, we did not find significant associations with any participant factors.

The following text has been added to the Methods:

A multivariable linear regression was used to compare the maximum post-baseline mutation load, transition/transversion ratio and spike escape in each participant. The covariates in the model were: treatment group, age, sex, log₁₀ baseline viral load, log₁₀ baseline spike antibody and the comorbidities: lung, heart, kidney, liver, neurological and immune disease, being a transplant recipient, obesity and hypertension.

The following text has been added to the Results:

In the multivariable model only treatment group was significantly associated with maximum mutation load, with molnupiravir being associated with on average 180 more mutations than Usual Care ($p < 0.001$). Similarly only treatment group was associated with maximum transition to transversion ratio, with molnupiravir being associated with an average ratio of 9.6 whereas the ratio in Usual Care was 5.3 ($p < 0.001$). No significant associations were found with the appearance of spike escape mutations.

- **As stated , perhaps MOL should be given longer but needs to be rigorously tested in a clinical trial and not just in a modeling study. The current study does not assess this so such broad claims cannot be made. Please temper this language**

We thank the reviewer for this comment. The conclusion in our abstract reads:

“Longer treatment courses should be tested...”

And our original conclusion read:

“These findings should be carefully considered and used to inform antiviral treatment strategies and future trials.”

To further temper the language in the conclusion it now reads:

These findings imply a need for a longer course of molnupiravir to be tested in a clinical trial and should inform decisions on whether to use the current 5 day course of molnupiravir until the results of such a trial are available.

Minor comments

- **Discussion could be shortened**

We have tried where possible to shorten the discussion whilst adding points raised by the reviewers. The revised manuscript remains within the word limit of 4000.

• **In the discussion of antibody blunting should cite this paper: Neutralizing Antibody Responses After Severe Acute Respiratory Syndrome Coronavirus 2 BA.2 and BA.2.12.1 Infection Do Not Neutralize BA.4 and BA.5 and Can Be Blunted by Nirmatrelvir/Ritonavir Treatment - PubMed (nih.gov)**

• **Similarly, it would be good to look at neutralization of antibodies after treatment, like the Carlin et al paper. If not, then at least note this in the limitations of the paper.**

We thank the reviewer for highlighting this paper which is now cited. Unfortunately we were unable to generate sufficient virus to undertake neutralisation assays as described, and did mention this in our limitations paragraph (original text: We were unable to confirm escape of mutated viruses from the effects of host antibodies through viral neutralisation as culture was unsuccessful for these samples.) The following text added to the discussion

Whilst we were unable to culture sufficient virus to undertake antibody neutralisation assays, the magnitude of difference in antibody titre between the arms by Day 14 (2200 U/mL lower in the molnupiravir arm) when antibody response usually peaks, is likely to be clinically significant. With populations becoming increasingly reliant on natural infection to boost immunity, antiviral blunting of antibody response is a concern [Ref to Carlin et al added].

Reviewer #2 (Remarks to the Author):

This is an interesting and very important study concerning the inadequacy of a five-day course of treatment of outpatient COVID-19 with molnupiravir, both from a treatment perspective and from the perspective of risk of transmission of resistant virus. I have the following comments followed by some specific details that it would be useful to address:

We thank the reviewer for this positive statement and provide point-by-point responses below.

1. The difference between arms in number of participants agreeing to provide additional less intensive sampling for the virology sub-study seems surprisingly large for a randomized trial (315 for Usual Care versus 248 for Molnupiravir). If agreement to participate was obtained prior to randomization then it would seem to be an unlucky chance imbalance but if agreement was obtained after randomization then it raises the possibility of bias in comparisons between arms beyond the factors that you adjusted for. Related to this:

a. Please clarify whether agreement was obtained before or after randomization and, if after, was it before or after a participant was made aware of their randomized assignment? (Extended data fig 1 seems to imply that it was after randomization so suggest reworking that figure if agreement was obtained before randomization).

We thank the reviewer for this comment and agree the figure is misleading. Invitation to participate in the virology sub-study was indeed done BEFORE randomisation. This imbalance seems to be an unlucky chance because of the original 6127 approached, 3093 (50.5%) went on to be randomised to Usual Care and 3034 (49.5%) to molnupiravir.

Reassuringly the proportion returning samples (90% versus 88%) was not significantly different between the groups ($p=0.2706$).

The flow chart has now been updated. As an aside, we noticed a further typographical error (as highlighted by the reviewer below) in the figure in that 6127 not 6131 were approached – this has been corrected in the figure.

b. If agreement was after randomization, then the limitations section should recognize the possibility of confounding due to factors, measured or unmeasured, beyond those that you adjusted for.

See comment above, this has now been clarified. Agreement was pre randomisation.

c. A suggestion is to give the proportions who returned samples by arm in lines 140 to 143, to establish that these are reasonably similar and unlikely a source of bias.

As above we have now added the following text:

Reassuringly the proportion returning samples (90% versus 88%) was not significantly different between the groups ($p=0.2706$).

2. Figure 2a is key for major conclusions but has the limitation that large numbers of measurements below the LLOQ are being imputed using the value LLOQ/2. A better presentation would be to plot the medians (alternatively, if assumptions are satisfied, use regression for censored data to estimate the mean at each time point).

We would argue the key plots are Figures 2b and c pertaining to the viral dynamic model which accounts for all viral loads (above and below LOQ) and corrects for important covariates rather than Figure 2a, which is a plot of the raw data with pairwise comparisons. As the reviewer points out, at later time points this comparison is complicated by proportion of <LLOQ but the numerical values of these proportions are presented for the reader to see this.

Nevertheless we take the reviewer's point regarding medians being more appropriate for the continuous part of the plot and have replaced Figure 2a which shows a very similar trend as the means but the central tendency of Usual Care drops below the LOQ at Day 14, which is now in line with the observed percentage <LLOQ for Usual Care.

Please also clarify why the CIs at Day 4 or narrower than those at Day 5 despite what should be a much larger sample size at Day 5 (or did “less intensive” participants often test a day early, in which case might they also have stopped taking treatment a day early? – what were the specific instructions for Day 5 testing?).

We would like to congratulate the reviewer for spotting this! It arose from a confusion in the definition of “Day” whereby the first batch of kits asked less intensive participants to send their antibody and second viral swab on the 4th day which would be “Day 5” if counting the first day as Day 0.

Because the study was done with patient self-sampling we always anticipated some early/late samples so the protocol had a time window of +/- 1 day for the post-baseline

samples from the outset. Because the kits were so expensive and it would have significantly delayed the study in the context of a very short recruitment window, we therefore used these kits and as a result did have a slightly higher than expected proportion of Day 4 samples which came from the less intensive group. This does not mean participants stopped taking their medication early and was not biased towards one arm or the other.

When fitting the viral and antibody dynamic models, and when presenting the Figure 2a raw data, we use the actual day. Since the models were fitted to all data with actual day of sampling used this would not have affected the modelling result.

3. The primary publication for the trial mentions participant-reported adherence information. It would be good to include a summary of that for the substudy population to establish whether or not participant-reported adherence was very high. The interpretation of results would need to be more cautious if participant-reported adherence was not very high.

This is an important point that we did not fully address. Participant-reported adherence was very high (96% reporting taking doses on all 5 days). The following text has been added and the analyses re-run to be “as treated” with the drug effect re-estimated as a fraction of the number of days participants reported taking their medication. Minimal differences were seen on parameter estimates as expected from such a small non-adherent population.

Participants in the molnupiravir arm were asked how many days of the 5 day treatment course they completed. Only 10 participants reported not taking the full course equating to 4% of those in the treated arm. Of these, 5 participants reporting not taking any, two only 1 day and one each reporting 2, 3 or 4 days of treatment.

4. A simulation study is used to estimate the probability of viral load being <109 copies/mL if molnupiravir was continued after 5 days. Please state in lines 173-4 the model being used to simulate this (presumably it is an extrapolation of the model over Days 1 to 5?). It should be made clear here and in the Abstract and Discussion that this is an extrapolation which cannot be validated based on data from this study. Please also detail to line 585-591 to clarify what was done.

We thank the reviewer for this comment and agree it requires clarification. The wording has now been updated as follows:

Results:

Extrapolation using simulated parameters from the viral dynamic model but with the drug effect extended suggest that to achieve a 95% probability of virus falling to less than the lower limit of quantification (<LLOQ), at least 10 days of therapy is required (Figure 2c).

Methods:

A simulated dataset of 1000 participants with demographics sampled from the virology sub-study demographic database was created. Each simulated participant was assigned model parameters based on their demographics and sampling from the parameter distribution. The drug effect was increased in daily increments with successive simulations

and the probability of virus being below the LLOQ reported for each treatment duration. This was repeated 1000 times to generate 95% prediction intervals.

5. Line 326-327: Please provide some additional underpinning to this conclusion. It suggests that most of the hospitalizations/deaths were after the end of treatment so it would be good to show cumulative incidence curves for hospitalization/death to confirm this.

We thank the reviewer for this comment but note that there were only 5 hospitalisations/deaths in the whole cohort (median 14 days, range 9-22). The statement on Line 326 was a discussion point rather than conclusion and with such small numbers we believe cumulative incidence curves adding a further plot would not be informative.

6. Discussion: Suggest contrasting your findings concerning predictors of baseline viral load and longitudinal changes with that of Moser et al (Open Forum Infect Dis, 2022), particularly regarding findings for sex differences.

Moser et al found age associated with baseline and Female participants to have faster viral decline. We found age and sex impacted on baseline viral load but not on decline rate. These findings are broadly in line with younger people and female participants having lower viral loads. Since there seemed to be more heterogeneity in time since symptom onset in the Moser study which will affect baseline if correlated with time since symptom onset (correlations not reported), this may have shown up in viral load. Our findings are therefore broadly similar but note neither study set out to look for these effects. We have added the citation.

Specific details:

Throughout: Some assays detect virus below the LLOQ. So please clarify whether “undetectable” means target not detected or whether it means below the LLOQ. If the latter, suggest using, e.g., “<LLOQ” rather than “undetectable”. Similarly, “viral clearance” in lines 173/4 should be changed to “<LLOQ” or “<109 copies/mL” as used in the legend of Figure 2).

This is an important point that we did not report clearly enough. Our assay calibration curve was linear to a Ct value of 40 equating to 109 copies/mL. We are therefore reporting limit of quantification rather than limit of detection.

All instances of “undetectable” have now been replaced with <LLOQ

Abstract: Please add that these are non-hospitalized adult participants.

Thank-you, this has been done.

Line 182 and Figure 2f: Based on the units for the AUC (copies*days/mL), it seems that the AUC is being calculated using the unlogged viral loads and hence is likely driven by the Day 1 level and so initial virus exposure---perhaps spike antibody time is similarly correlated with Day 1 viral load? It would be good to check and clarify this. Also clarify whether AUC is model-based, or based on actual measurements.

This is a good point. Using the trapezoidal rule we should have calculated AUC on the log transformed concentrations given the exponential decay. We have now corrected this

(Figure 2f shows a very similar trend) and we now clarify the AUC was calculated using individual model parameter predictions at fixed time points given heterogeneity in timing of some sample returns.

Regarding the correlation with antibody baseline and slope, this is an excellent point that we did not address. The correlation coefficient calculated from the variance covariance matrix between antibody baseline and slope was -0.632, and hence the lower the antibody titre at baseline (and therefore higher initial viral load), the higher the rate of increase.

But viral load at baseline alone cannot be the sole driver of antibody increase rate, as viral load baselines were similar between the groups. However, molnupiravir was significantly associated with slower antibody increase, which we believe relates to the early rapid decline in viral load, so we still maintain that the viral AUC versus antibody slope gives important insight.

We have added the following text:

The viral and antibody models were also used to extract individual parameter estimates for each participant. These were then used to calculate the area under the curve (AUC) from Day 0-14 using the trapezoidal rule on Log10 transformed individual model predictions of viral load and compared with individual predicted spike antibody doubling times.

Line 205: Is the substantial difference in rate of successful sequencing between arms due to the reasonably modest difference in viral load shown in Figure 2a? It would be good to explain the difference to reassure the reader that other factors are not causing the difference.

We attempted to sequence all samples with Ct values <38. Viral sequencing is unlikely to be successful in samples with lower viral load, and although by Day 14 these differences are modest, they are at the borderline of what is possible, so this is the likely explanation. We have updated the text as follows:

Of 145 participants with viral Ct <38 at Day 14 (nine days following treatment end), 46/75 in the molnupiravir and 21/70 in the Usual Care group were successfully sequenced, the higher success rates in the molnupiravir samples likely due to higher viral loads.

Line 213: why 45 and not 46 as in line 205?

Thank you, this should indeed read 46 and we have corrected the text.

Lines 309-10 and 171-172: This is a key finding, presumably based on 44% for Usual Care and 52% for Molnupiravir being above LLOQ? It would be good to be explicit about this and that there is a significant increase at Day 14.

The statement on Line 309 is not only derived from percentage <LLOQ at a single time point. The picture is much more nuanced as we talk about significant initial decrease followed by significant slowing in viral decline post-treatment. On line 171 we already state: "By Day 14 viral loads were significantly higher in molnupiravir-treated participants than Usual Care."

Line 317: Add “copies/mL” before “difference”.

This has been added

Line 493: It seems that this sentence is about the intensive sampling. If so, it would be good to clarify that.

Clarified in text.

Lines 527 to 533: What was the upper limit of quantification for this assay and how were results above this handled in the modeling? (or was there retesting of diluted samples if an initial result was above an upper limit?).

The upper limit of the viral load calibration curve was 10^{10} copies/mL. Our highest viral load was $10^{9.6}$ so no dilutions or censoring the model in the upper direction was needed.

Line 563: See above comment about avoiding using imputed values in the analysis. Also what were the categorical variables being tested?

This has now been revised as per major comment on Figure 2a. The categorical variables are those listed in Extended Table 1. This text from methods is deleted as it already appears in the table legend.

Pairwise comparisons of the median of viral load were compared at each time point using a Mann-Whitney test with samples below the limit of quantification being replaced with the LLOQ/2.

Line 567-9: Please confirm that the linear model provided adequate fit in this specific study.

We agree with the reviewer that model fit is important and that was addressed in our original submission. In the supplementary material, Figures 1a and b show plots of predictions versus observations and standardised residuals for the viral and antibody dynamic models respectively and Figures 2a and b show individual model fits for each arm respectively. Whilst “all models are wrong”, these diagnostic plots clearly show adequate fit to describe the data.

Line 570: “probability” might be better than “likelihood”.

Thank you this has been changed.

Lines 573-4: Please clarify what you mean by “on baseline viral load and slope”.

The basic model contains two parameters, a baseline viral load (intercept) and decline slope. Covariates were tested to affect each in turn. As per changes to the model description in other responses this should now be more explicit.

Line 576: i should be subscript in “pi”.

Thank you this has been corrected.

Line 611: Please clarify what you mean by “were adjusted for viral load”.

Thank you, this is an error: it was unnecessary to adjust for viral load and this has now been removed.

Figure 1 legend: Note your criterion for selecting when to show p-values.

Added

Extended data tables 2 and 3: (a) Add units for all parameters (and give correct units for delta). (b) “Additive error variance” might be clearer, with units stated also. (c) Add an algebraic definition of the model being described (this likely will help some readers understand the parametrization).

The correct units for delta are the reciprocal of time so the units presented are correct. We assume the reviewer prefers a different notation so this has been replaced with d^{-1} . The coefficients are dimensionless. As per queries from Reviewer 3 (see below) parameterisation and interpretation of covariates are now added.

Extended data figure 1: (a) You have $n=6131$ splitting into $n=3093+3034=6127$. Please correct/clarify (line 134 also mentions 6127 so perhaps that is incorrect also?). (b) Add an additional row of boxes at the bottom showing the number included in the analysis (this will also clarify whether or not the exclusions are included in the count of participants who returned samples).

See response above, 6127 is the correct number and the flow chart figure is now corrected.

Extended data figure 6: The legend mentions $n=558$ and $n=425$ “participants” but these are larger than the study population. Please clarify. If it is counting multiple samples per participant, then how did you address this in statistical analysis?

Thank you this is a typographical error and should read “samples with sufficient viral load” rather than “participants”. This has been corrected.

Supplementary Figure 1a: I’m unclear what are the DV values shown when they are block dots and so below LLOQ?

The black dots relate to imputed values of the below limit of quantification samples used for graphical diagnostic purposes only when modelling $<LLOQ$ data. These are standard outputs in nlmixr2 and related software. Interested readers will be able to trace back their origins via software documentation to:

Nguyen T. J Pharmacokinet Pharmacodyn (2012) 39:499–518 DOI 10.1007/s10928-012-9264-2

Reviewer #3 (Remarks to the Author):

Standing et al. performed detailed analysis of SARS-CoV-2 dynamics and evolution in molnupiravir-treated individuals, with several key findings: 5 days of molnupiravir reduces viral load but does not result in virus clearance; it appears to blunt the antibody response; and there are mutated but replication-competent viruses remaining at 2 weeks after treatment initiation. Overall, this is a really well-done and well-described study that adds valuable insight into the use of molnupiravir treatment for SARS-CoV-2. I have a few questions and comments:

We thank the reviewer for this positive statement and provide point-by-point responses below.

1. Please provide more details about the multivariable model that was used to evaluate parameters associated with viral load decline (lines 166-171). The beta coefficients from extended data 2 should be discussed in the main text to help the reader understand the magnitude of change.

This is an important comment, we have not explained this in sufficient detail. In revising the manuscript we have taken the opportunity to change to an as-treated analysis for the viral dynamic and antibody models (which is in line with our SAP). Parameter estimates are very similar to our original submission.

The following text has been added to aid interpretation of the model output:

The model predicts male participants have on average 0.2 log₁₀ copies/mL higher baseline viral load whereas for a 10 year increase in age from the median a 0.13 log₁₀ copies/mL increase in viral load is expected. A 0.5 log₁₀ U/mL decrease in spike antibody from the population median is associated with a 0.18 log₁₀ copies/mL increase in viral load. Time since symptom onset increasing from 2 to 5 days is associated with a 0.9 log₁₀ copies/mL decrease in viral load. The viral decline half-life in Usual Care was 0.72 days. This decreased to 0.41 days when on molnupiravir, and then increased to 1.71 days post-treatment.

In extended data 2, it also needs to be more clear which beta coefficients affected baseline viral load vs. slope; it took several reads of the legend and I'm still not sure I understand this. Suggest adding another column or horizontal breaks in the table to indicate which parameter of the model was affected by each variable.

Rather than amending the table, as suggested by Reviewer 2 the model is now written out in the legend.

The slope, δ , is the first order decline rate in viral load, whereas the intercept ($V(0)$) the baseline viral load. Inter-individual variability (IIV) was assumed to follow a log-normal distribution for δ and log₁₀ for $V(0)$. Additive error is the variance of the residual error term. The following covariates all significantly ($p < 0.01$) improved model fit: β_{mol_on} refers to the fractional change in delta when on molnupiravir, β_{mol_off} refers to the fractional change in delta in the post-molnupiravir treatment period. β_{sex} is the change in baseline viral load in males. β_{age} is the allometric exponent relating age with $V(0)$. β_{ab} is the allometric exponent for the inverse relationship of $V(0)$ load and spike antibody. β_{sym} is the allometric exponent

for the inverse relationship between $V(0)$ and time since symptom onset. The final model describing viral load with time ($V(t)$) was therefore:

$$V(t) = V(0) \beta_{\text{sex}} I_{\text{male}} (\text{age}_i / \text{age}_{\text{median}})^{\beta_{\text{age}}} (A(0)_i / A(0)_{\text{median}})^{\beta_{\text{ab}}} (\text{tsym}_i / \text{tsym}_{\text{median}})^{\beta_{\text{sym}}} \exp\{-\delta t I_{\text{mol1}} \beta_{\text{mol_on}} I_{\text{mol2}} \beta_{\text{mol_off}}\},$$

where I_{male} is an indicator for male participants; age , $A(0)$ and tsym are the individual and population median age in years, log10 spike antibody baseline and tsym time since symptom onset respectively; I_{mol1} and I_{mol2} are time varying indicators for the molnupiravir arm during and post treatment.

Same comments for the analysis of spike antibody concentrations (lines 176-184).

As with the viral model, the antibody parameters are more explicitly stated:

Table legend:

The slope, δ , is the first order increase rate in spike antibody, whereas the intercept ($A(0)$) the baseline spike antibody level. Inter-individual variability (IIV) was assumed to follow a log-normal distribution and estimated for δ and $A(0)$. Additive error is the variance of the residual error term. The following covariates all significantly ($p < 0.01$) improved model fit: β_{sex} is the fractional change in $A(0)$ in males. β_{vac} is the fractional change in $A(0)$ in participants who were not fully vaccinated. β_{mol} is the fractional change in δ in participants receiving molnupiravir. The final model describing antibody with time ($A(t)$) was therefore:

$$A(t) = A(0) \beta_{\text{sex}} I_{\text{male}} \beta_{\text{vac}} I_{\text{unvacc}} \exp\{-\delta t I_{\text{mol}} \beta_{\text{mol}}\},$$

where I_{male} is an indicator for male participants, I_{unvacc} is an indicator for participants who were not fully vaccinated, and I_{mol} is an indicator for participants who took molnupiravir.

2. Please describe how simulations of the viral dynamic model were performed (lines 172-174). I don't see this in the methods.

Thank you for spotting this omission. Now added as per response to reviewer 2 above.

3. Using a minimum cutoff of 10X sequencing depth (line 188) does not square with analyzing SNPs down to 1% frequency (extended data figure 3), which would require over 100X depth. I am also not convinced that read depth and SNP detection aren't associated. It's not clear why the bins in Supplementary Figure 4 were set the way they are (<250, 250-500, 500-750), but a scatter plot with correlation analysis would be more convincing.

We thank the reviewer for this comment. As stated in the methods section, the target depth was on average around 5000X per genome (315X-12318X) allowing analysis of minor variant calling down to 1%. Allele frequency (AF) of 1% was mainly used to calculate the total number of mutations, while for other analyses involving minor variants, we used a minimum AF $\geq 5\%$ and consensus level allele frequency (AF > 50%). The 10X cut-off was used for consensus level.

We have now done as the reviewer suggests and replaced the Supplementary Figure 4 with a scatter plot showing no correlation between SNP detection and mean read depth.

4. The statement that “emerging spike antibody mutations... were possibly associated with slower decline” (lines 239-241) should include a result of statistical analysis, because the slopes in Figure 5b don’t appear different. Additionally, total number of mutations is an important confounder in this analysis.

We agree with the reviewer here, indeed the decline rate is not statistically significantly different. This statement cannot be supported and has now been removed.

5. The statement that “we did not find a major difference in the proportion with recoverable virus between groups” (line 296) should include results of a statistical analysis.

Thank you. This has now been addressed in the figure and text in response to Reviewer 1.

6. Please modify the discussion about the duration of treatment in other infectious diseases (lines 324-326, 462-464); while I agree with the authors that their study supports a longer treatment course for molnupiravir, we almost never perform tests of clearance before deciding to stop antimicrobial therapy for other infections and I doubt we would do so for SARS-CoV-2.

We are not advocating tests for clearance before deciding to stop. On neither lines 324 or 462 do we advocate performing tests for clearance for SARS-CoV-2, although there are many infectious diseases for which personalised treatment duration/choice is used via pathogen clearance tests: e.g. antibiotics would not be stopped in sepsis if blood cultures remained positive.

We therefore stand by our wording which states that, as in other infectious diseases where dose recommendations are made based on clinical trial results, antiviral courses should be long enough to clear the pathogen. This remains a fundamental principle of antimicrobial chemotherapy. Course length is determined in clinical trials (usually Phase II), and should be sufficient to ensure pathogen clearance in the majority of patients. Our clinical trial result clearly shows that 5 days is insufficient to clear the virus for the majority of patients treated with molnupiravir. We are not aware of another infectious disease where antimicrobial course length is so short that the pathogen is not cleared in the majority of patients.

REVIEWERS' COMMENTS

Reviewer #2 (Remarks to the Author):

None

Reviewer #3 (Remarks to the Author):

The authors have addressed my prior comments well. Congratulations on a really nice study.